



# Chemical ionization mass spectrometry utilizing benzene cations for measurements of volatile organic compounds and nitric oxide

Uma Puttu[1], Jamie R. Kamp[1], Xiaoyu Chen[1], Jhao-Hong Chen[1], Jing Li[1], Miquel A Gonzalez-Meler[2], Jian Wang[1], and Lu Xu[1]

[1]Department of Energy, Environmental, and Chemical Engineering, Washington University in St. Louis, St. Louis, MO, USA
[2]Department of Biological Sciences, University of Illinois at Chicago, Chicago, IL, USA

**Correspondence:** Lu Xu (xu1@wustl.edu)

**Abstract.**

We evaluate the capability of chemical ionization mass spectrometry (CIMS) using benzene cations as reagent ions (benzene CIMS) for detecting atmospheric trace gases. We characterize the ionization pathways and product ion distributions for 27

analytes spanning diverse chemical classes. To interpret the complex ion chemistry involving two reagent ions ($C_6H_6^+$ and $(C_6H_6)_2^+$) and multiple ionization pathways (charge transfer, proton transfer, adduct formation, and hydride abstraction), we introduce a thermodynamics-based framework that classifies analytes into three categories based on their ionization energy (IE), relative to those of benzene monomer (9.24 eV) and dimer (8.69 eV). Each class exhibits distinct ionization mechanisms and product ions. Analytes with IE smaller than 8.69 eV (low IE class) undergo charge transfer with both reagent ions; analytes

with IE between 8.69 and 9.24 eV (mid IE class) undergo charge transfer with $C_6H_6^+$ and potential adduct formation with $(C_6H_6)_2^+$; analytes with IE larger than 9.24 eV (high IE class) could undergo adduct formation, proton transfer, or hydride abstraction. Analytes within each class also show similar sensitivity, enabling sensitivity estimation for compounds lacking calibration standards. In addition to volatile organic compounds (VOCs), benzene CIMS detects nitric oxide (NO) with a 1-minute detection limit of 5 pptv, exceeding the performance of most commercial $NO_x$ analyzers. Field deployments in Chicago

and St. Louis demonstrate good agreement with reference NO measurements. Isoprene measurements show good agreement with a co-located gas chromatography–photoionization detector (GC-PID) in St. Louis, but exhibit substantial positive bias in Chicago, likely due to interferences from anthropogenic VOCs in the polluted urban environment. These results highlight the potential of benzene CIMS for concurrent measurements of NO, VOCs, and their oxidation products using a single instrument, while also underscoring challenges in complex atmospheric conditions.

# 1    Introduction

The chemical reactions between volatile organic compounds (VOCs) and nitrogen oxides ($NO_x$, sum of NO and $NO_2$) play a central role in atmospheric chemistry, as they produce air pollutants, such as ozone ($O_3$), particulate matter, and influence the oxidizing capability of the atmosphere (Ziemann and Atkinson, 2012). These reactions are initiated by oxidation of VOCs by atmospheric oxidants such as hydroxyl radical (OH), $O_3$, and nitrate radical ($NO_3$), which leads to the formation of organic



peroxy radicals ($RO_2$). The subsequent fate of $RO_2$ radicals depends strongly on the concentration of nitric oxide (NO), resulting in different oxidation products (Orlando and Tyndall, 2012).

VOCs originate from biogenic, pyrogenic, and anthropogenic sources (Guenther et al., 1995, 2012). Globally, biogenic sources dominate VOC emissions. Isoprene has an estimated emission rate of around 500 TgC per year, accounting for 70% of biogenic VOCs (Guenther et al., 2006, 2012). Pyrogenic sources, including wildfires and prescribed burns, are the second-

largest global source of nonmethane organic gases (Yokelson et al., 2008). The primary sources of anthropogenic emissions include vehicular emissions, industrial processes, and volatile chemical products (VCPs) (McDonald et al., 2018; Wallington et al., 2022). On the other hand, $NO_x$ is produced from fossil fuel combustion and natural sources such as soils, lightning, and open fires (Adams et al., 2019; Lasek and Lajnert, 2022). Given the importance of VOCs and $NO_x$, extensive efforts have been made to measure them. However, their measurements are typically conducted using different instruments.

One widely used method for measuring VOCs is gas chromatography (GC), coupled with detectors such as photoionization detection, flame ionization detection, and mass spectrometry. The GC technique offers high sensitivity, allowing for the detection of numerous VOCs with volume mixing ratios as low as 0.1 pptv (parts-per-trillion by volume). However, GC typically requires several minutes of sampling to collect sufficient material, followed by additional time to elute analytes through the column. This relatively slow response limits its ability to capture the rapid temporal variability of atmospheric compounds.

Alternatively, chemical ionization mass spectrometry (CIMS) is widely used to detect VOCs at high temporal resolution, achieving measurement rates as rapid as 10Hz (Bertram et al., 2011). CIMS selectively detects target species by ionizing them with reagent ions. Hydronium ion ($H_3O^+$) is the most widely used reagent ion for measuring VOCs (de Gouw et al., 2003; Yuan et al., 2016). However, one challenge in this technique is measurement interferences. For instance, the $H_3O^+$ ion chemistry can break some aldehydes and ketones into smaller fragments that overlap with isoprene signals, complicating its quantification

(Coggon et al., 2024). Other reagent ions such as $NO^+$, $O_2^+$, $C_6H_6^+$, $(CH_3COCH_3)H^+$ and $NH_4^+$ have been explored to detect VOCs (Spanel and Smith, 1997; Koss et al., 2016; Dong et al., 2022; Xu et al., 2022).

NO is typically measured using chemiluminescence (CL) and laser-induced fluorescence (LIF) techniques. The CL method quantifies NO by measuring the intensity of chemiluminescence emitted when NO reacts with excess $O_3$. This method is widely used because of its straightforward operation and linear response to a wide range of NO concentrations (Fontijn et al.,

1970; Ridley and Howlett, 1974; Steffenson and Stedman, 1974; Kley and McFarland, 1980; Bollinger, 1982). Commercial NO analyzers typically have detection limits on the order of 100 pptv, limited by background noise. By reducing background count rates, research-grade instruments can further improve detection limits to 0.5 - 1 pptv (Ridley and Grahek, 1990). The LIF method quantifies NO by exciting NO molecules at 226 nm and detecting the emitted fluorescence as they return to their lower energy state (Bradshaw et al., 1982). Detection limits of the LIF method have improved to 1 pptv and 0.3 pptv for a 1

s and 10 s integration times, respectively. The measurement uncertainties are reduced to below 0.2 pptv (Rollins et al., 2020). One limitation of $NO_x$ analyzers based on either CL or LIF technique is that they are only capable of measuring one to three species at maximum (NO, $NO_2$, and $NO_y$).

Simultaneous measurement of both VOCs and NO using a single instrument would be highly advantageous, as it reduces instrumentation complexity and ensures collocated, time-resolved data for key reactive species. Here, we explore the capability





of CIMS using benzene cations as reagent ions (here after referred to as benzene CIMS) for this purpose. Benzene CIMS has been previously applied to detect various VOCs, including isoprene, monoterpenes, and dimethyl sulfide (DMS) (Leibrock et al., 2003; Kim et al., 2016; Lavi et al., 2018). Some evidence in the literature suggests that benzene cations may also react with NO. For example, Sieck and Gorden (1976) reported the formation of benzene–NO clusters ($C_6H_6NO^+$) at 3 mbar. Additionally, Mungall et al. (2016) observed $C_6H_6NO^+$ peaks while deploying benzene CIMS in the Arctic, though they attributed

these signals to impurities in the zero air, which was used as the carrier gas for reagent gas. These observations motivated us to investigate the feasibility of using benzene CIMS for the concurrent detection of VOCs and NO in the atmosphere.

Recent advances in CIMS have facilitated the broader use of benzene cations as reagent ions. Ji et al. (2020) developed a novel ion source that employed vacuum ultraviolet (VUV) light to generate benzene cations. This VUV source can also produce negative reagent ions such as $I^-$ and $NO_3^-$ (Riva et al., 2024). This enables rapid switching between reagent ions of

opposite polarities within a single CIMS. Because these reagent ions are sensitive to compounds with different chemical functionalities, they provide complementary detection capabilities, making a single instrument capable for characterizing complex atmospheric mixtures. While the ion chemistry of $I^-$ and $NO_3^-$ has been extensively studied (Jokinen et al., 2012; Lee et al., 2014), benzene ion chemistry remains less explored. Laboratory studies have characterized benzene CIMS for detecting isoprene, monoterpenes, DMS, and ammonia ($NH_3$) (Leibrock and Huey, 2000; Kim et al., 2016; Mungall et al., 2016; Lavi et al.,

2018; Schobesberger et al., 2023), but a comprehensive evaluation of its broader measurement capabilities is still lacking. Field deployments of benzene CIMS are also limited. For example, Leibrock et al. (2003) used benzene CIMS to measure isoprene in Boulder, Colorado in 2003, and reported concentrations higher than those measured by co-located GC measurements. They attributed the difference to interferences from unidentified anthropogenic compounds. Since then, benzene CIMS has not been evaluated for quantifying isoprene in the atmosphere.

In this study, we systematically investigate benzene cation chemistry and evaluate its detection capability for a broad range of atmospheric species. In the laboratory, we characterize the product ion distribution, sensitivity, and the response of analytes to key instrument operating conditions for a number of analytes spanning several chemical functional classes. By combining these laboratory results with findings from previous studies, we develop a framework to assess detection capabilities, predict product ions, and estimate sensitivities. Finally, we deploy the benzene CIMS at two field sites to evaluate its performance

under ambient conditions.

## 2 Instrument description

### 2.1 Chemical ionization mass spectrometer (CIMS)

The CIMS used in this work is a Tofwerk Vocus CI-TOF 2R equipped with an Adduct Ionization Mechanism (AIM) Ion-Molecule Reactor (IMR). Reagent ions are generated by photoionizing benzene using a VUV source (UV lamp krypton DC

PID PKS 106, Heraeus). Ultra-high purity (UHP) nitrogen ($N_2$) gas flows over two permeation tubes containing liquid benzene (Sigma Aldrich, $\geq$ 99.9%) maintained at 80°C in a compact oven, carrying benzene vapor into the VUV source. The UV lamp emits photons at 124 nm and 117 nm wavelengths, corresponding to energies of 10 and 10.6 eV, respectively. These





photons ionize benzene, which has an ionization energy (IE) of 9.24 eV, forming benzene monomer cations ($C_6H_6^+$) (reaction R1) (Ji et al., 2020; Breitenlechner et al., 2022). These $C_6H_6^+$ ions can further react with benzene in the IMR to form benzene dimer cations (($C_6H_6)_2^+$) (reaction R2). The observed mass spectrum peak ratio of $C_6H_6^+/(C_6H_6)_2^+$ in our instrument is about 5. Additional clustering in IMR can lead to the formation of larger benzene cluster ions ($C_6H_6)_n^+$ with (n $\geq$ 3) (Grover et al., 1987). However, such larger clusters are not detected in our instrument, likely because they are weakly bound and decluster in the ion optics (Kim et al., 2016; Lavi et al., 2018).

$$C_6H_6 + h\nu \rightarrow C_6H_6^+ + e^- \tag{R1}$$

$$C_6H_6^+ + C_6H_6 \rightleftharpoons (C_6H_6)_2^+ \tag{R2}$$

The reagent ions $C_6H_6^+$ and ($C_6H_6)_2^+$ react with analytes in the IMR. Sample flow is introduced into the IMR at a rate of 1.8 standard liters per minute (slpm), while reagent ions are introduced at a 45° angle to optimize intersection with the sample flow and minimize deflection. The IMR is a conical reaction chamber composed of conductive polytetrafluoroethylene, which minimizes vapor-wall interactions and suppresses turbulent eddies, thereby enhancing equilibration speed and reducing memory effects. The residence time in the IMR is approximately 10 milliseconds. Further details about the IMR can be found in Riva et al. (2024). After exiting the IMR, product ions pass through a critical orifice at a flow rate of 0.5 slpm, then sequentially traverse a small-segmented quadrupole (SSQ), a big-segmented quadrupole (BSQ), a series of DC optics before entering a time-of-flight (ToF) mass analyzer. The ToF analyzer has a mass resolution ($m/\Delta m$) of 9000, enabling the separation of isobaric ions.

## 2.2 Benzene ion chemistry

The system contains two reagent ions (denoted as $R^+$), $C_6H_6^+$ and ($C_6H_6)_2^+$. Each reagent ion could have multiple reaction pathways with analytes (denoted as X), including charge transfer, adduct formation, proton transfer, and hydride abstraction. Below, we first describe each of these reaction pathways, followed by a discussion of the competition among them.

Charge transfer occurs when the IE of the analyte is lower than that of the reagent ion (reaction R3). The IE values for $C_6H_6^+$ and ($C_6H_6)_2^+$ are 9.24 eV and 8.69 eV, respectively (Grover et al., 1987; Linstrom et al., 1997). The predominant product is the analyte molecular ion ($X^+$). Fragmentation may also occur via dissociative charge transfer (reaction R4), if the appearance energy (AE), which is the minimum energy required to produce a specific fragment ion from the analyte, is smaller than the IE of reagent ion (Gross, 2017).

$$R^+ + X \longrightarrow X^+ + R \tag{R3}$$

$$R^+ + X \longrightarrow [X-frag]^+ + R + frag, \quad frag=H, CH_3, \text{ or other fragments} \tag{R4}$$





Adduct formation ($C_6H_6 \cdot X^+$) can occur through two mechanisms: analyte clustering with $C_6H_6^+$ (reaction R5) and ligand exchange with $(C_6H_6)_2^+$ (reaction R6). A critical parameter governing adduct formation is the analyte's $C_6H_6^+$ affinity, defined as

the negative of the enthalpy change for the reaction between $C_6H_6^+$ and the analyte, analogous to proton affinity and $NH_4^+$ affinity (Xu et al., 2022). If the analyte has a higher $C_6H_6^+$ affinity than benzene (74 kJ mol$^{-1}$), both reaction pathways (R5 and R6) are thermodynamically favorable and lead to the formation of $C_6H_6 \cdot X^+$. Conversely, if the analyte has a lower $C_6H_6^+$ affinity than benzene, reaction R6 does not occur. Reaction R5 may still proceed if exothermic, but the resulting $C_6H_6 \cdot X^+$ is susceptible to ligand exchange by $C_6H_6$, forming $(C_6H_6)_2^+$ (reaction R7).

$$C_6H_6^+ + X \longrightarrow C_6H_6 \cdot X^+ \qquad\qquad\qquad\qquad (R5)$$

$$(C_6H_6)_2^+ + X \longrightarrow C_6H_6 \cdot X^+ + C_6H_6 \qquad\qquad\qquad\qquad (R6)$$

$$C_6H_6 \cdot X^+ + C_6H_6 \longrightarrow (C_6H_6)_2^+ + X \qquad\qquad\qquad\qquad (R7)$$

Proton transfer reaction occurs when the proton affinity (PA) of the analyte is higher than that of the phenyl radical (884 kJ

mol$^{-1}$) (Hunter and Lias, 1998), generating the protonated analyte ion $XH^+$ (reaction R8). Hydride abstraction reaction occurs when reaction R9 is exothermic, generating product ion $[X-H]^+$. In addition to these four major reaction pathways, some analytes can undergo secondary ion-molecule chemistry involving $O_2$.

$$C_6H_6^+ + X \longrightarrow XH^+ + C_6H_5^{\bullet} \qquad\qquad\qquad\qquad (R8)$$

$$C_6H_6^+ + X \longrightarrow [X-H]^+ + C_6H_7^{\bullet} \qquad\qquad\qquad\qquad (R9)$$

## 2.3 Analyte classification

Due to the presence of two reagent ions and multiple reaction pathways, analyte ionization involves competing pathways that depend on the analyte's properties (e.g., IE, PA, and $C_6H_6^+$ affinity) and the relative abundances of two reagent ions. To investigate these ionization pathways, we measured product ion distributions for 27 analytes. In these experiments, 10 sccm of $N_2$ was passed over a vial containing a small amount (<1 μL) of pure analyte, and the resulting vapor was diluted with 3 slpm

of $N_2$ before entering the instrument. Product ions were identified as those exhibiting a correlation coefficient ($R^2$) greater than 0.95 with the primary product ion and contributing more than 1% of its signal.

Similar competition has been studied for other reagent ions, such as $NO^+$. By investigating the chemistry of $NO^+$ with over 50 analytes, Koss et al. (2016) suggested that charge transfer dominates when it is thermodynamically favorable, regardless of





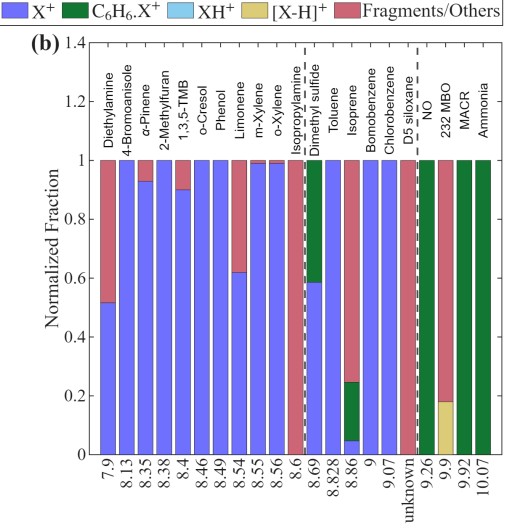

**Figure 1. (a)** Reaction pathways between reagent ions ($C_6H_6^+$ and $(C_6H_6)_2^+$) and analyte (X) across different analyte categories, along with the corresponding product ions. Symbols "✓", "✗", and "?" indicate high, low, and uncertain likelihood of the reaction, respectively. **(b)** Product ion distributions for each analyte, sorted by ionization energy (IE). The two vertical dashed lines mark the IE of 8.69 eV (benzene dimer) and 9.24 eV (benzene monomer). The IE of D5-siloxane is unknown, but likely between benzene dimer and benzene monomer, as discussed in the text. The same product ion formed via different reagent ions or ionization pathways are shown in the same color. Fragment ions from dissociative charge transfer and other reaction pathways are also shown in the same color, as they are difficult to distinguish experimentally.

other possible reactions. When charge transfer is not thermodynamically favorable, competition among other reaction pathways
needs to be considered. Building upon this principle, we categorize analytes based on their IE relative to benzene monomer and dimer, evaluate the competitive reaction pathways within each category (Figure 1a).

Analytes with IE smaller than those of both benzene dimer (8.69 eV) and benzene monomer (9.24 eV) are classified into "low IE class". We examine 11 analytes in this class and find that the charge transfer is the sole ionization pathway for all of them (Figure 1b). For 8 of the 11 analytes, the charge transfer product, i.e., analyte molecular ion $X^+$, accounts for more than
95% of total products. In the remaining three cases (limonene, isopropylamine (IPA), and diethylamine (DEA)), additional fragment ions $[X-H]^+$ or $[X-CH_3]^+$ are detected. These fragments arise from the dissociative charge transfer of $X^+$ (reaction R4), consistent with their appearance energies being lower than the IE of benzene (Table S2).

Several analytes in the low IE class have PA greater than that of the phenyl radical (884 kJ mol$^{-1}$), for example, IPA (923.8 kJ mol$^{-1}$) and DEA (952.4 kJ mol$^{-1}$). Despite the proton transfer is thermodynamically favorable for these analytes, their
protonated ions ($XH^+$) are not detected. This suggests these analytes preferentially undergo charge transfer rather than proton transfer. This observation is consistent with Allgood et al. (1990), who observed proton transfer ions only for analytes which could not undergo charge transfer (SI Section 1.2). Thus, for analytes in class low IE, charge transfer is the dominant ionization mechanism, regardless of other possible reactions.



Analytes with IE between that of benzene dimer and monomer are classified into "mid IE class". Their reactions with $C_6H_6^+$ and $(C_6H_6)_2^+$ are different. When reacting with $C_6H_6^+$, charge transfer dominates, the same as those in "low IE class". In contrast, charge transfer does not occur with $(C_6H_6)_2^+$, as the analyte' IEs exceeds that of benzene dimer. In such cases, other reaction pathways become possible. We examine six analytes in this class. For chlorobenzene, bromobenzene, and toluene, the molecular ion $X^+$ formed via charge transfer is the sole observed product. D5 siloxane also appears to belong to this class, though its IE is not directly known. Experimental observations suggest that its IE is between 8.83 eV (toluene) and 9.24 eV (benzene), because its molecular ion is not observed when using toluene cations as reagent ions (Alton and Browne, 2020) and because a fragmentation ion ($C_9H_{27}O_5Si_5^+$) is detected with benzene cations, presumably formed via dissociative charge transfer (reaction R4).

For DMS, both molecular ion ($C_2H_6S^+$) and adduct ion ($C_8H_{12}S^+$) are observed. The adduct formation is consistent with that the $C_6H_6^+$ affinity of DMS (113.05 kJ mol$^{-1}$ calculated by Vermeuel (2021)) is higher than that of benzene. The detection of both $C_2H_6S^+$ and $C_8H_{12}S^+$ in our study contrasts with previous studies by Mungall et al. (2016) and Schobesberger et al. (2023), who observed only $C_2H_6S^+$. This discrepancy is likely due to differences in electric field strengths. Kim et al. (2016) showed that a stronger electric field can decluster $C_8H_{12}S^+$ into $C_2H_6S^+$. Because the electric fields were not specified in previous studies, direct comparisons are difficult. In our instrument, the voltage gradient between the BSQ front and the skimmer is only 1 V. Ions are transported axially by the gas flow and focused radially by the radio frequency (RF) field, which minimizes added energy and thereby reduces declustering (Riva et al., 2024). In addition to declustering, another possible explanation for the formation of $C_2H_6S^+$ in previous studies is electric-field-assisted charge transfer between DMS and $(C_6H_6)_2^+$. Since DMS and $(C_6H_6)_2^+$ have nearly identical IE, the charge transfer reaction is borderline endothermic. A sufficiently strong electric field may provide the activation energy needed to overcome this barrier and produce $C_2H_6S^+$.

Isoprene exhibits a complex distribution of product ions (Figure 2). The distribution depends strongly on the type of dilution gas. When UHP $N_2$ is used, the dominant products are the adduct ion ($C_{11}H_{14}^+$, 55%) and the charge transfer ion ($C_5H_8^+$, 36%). In contrast, when zero air is the dilution gas, the fractions of both ions decrease and numerous oxygenated ions appear, including $C_4H_7O^+$, $C_5H_8O_2^+$, $C_4H_9O_2^+$, $C_3H_7O_2^+$, and $C_5H_6O^+$. The formation mechanisms of these oxygenated ions are uncertain but may involve $O_2^+$ chemistry or secondary ion–molecule reactions involving $O_2$. The complex distribution is not inconsistent with previous studies when differences in experimental conditions are considered. Schobesberger et al. (2023) reported $C_{11}H_{14}^+$ as the major product ion, but used $N_2$ as the dilution gas. Lavi et al. (2018) used zero air and also found $C_{11}H_{14}^+$ as the dominant product, but in that case $(C_6H_6)_2^+$ was the primary reagent ion, whereas $C_6H_6^+$ is the major reagent ion in our study. Charge transfer reaction with $C_6H_6^+$ is more energetic and leads to enhanced fragmentation and secondary ion chemistry compared to the adduct formation pathway of $(C_6H_6)_2^+$. It is also possible that $O_2^+$ chemistry played a smaller role in Lavi et al. (2018).

Analytes with IE greater than that of benzene monomer are classified into "high IE class". Due to their high IE, these analytes cannot undergo charge transfer with either reagent ion. Instead, their potential ionization pathways include proton transfer, adduct formation, and hydride transfer. The dominant pathway depends on the thermodynamic properties of each analyte. For NO, $NH_3$, and methacrolein (MACR), proton transfer is not thermodynamically favorable, as their PA are all lower than that of the phenyl radical (Table S1). Hydride abstraction is also unfavorable, because NO lacks a hydrogen atom





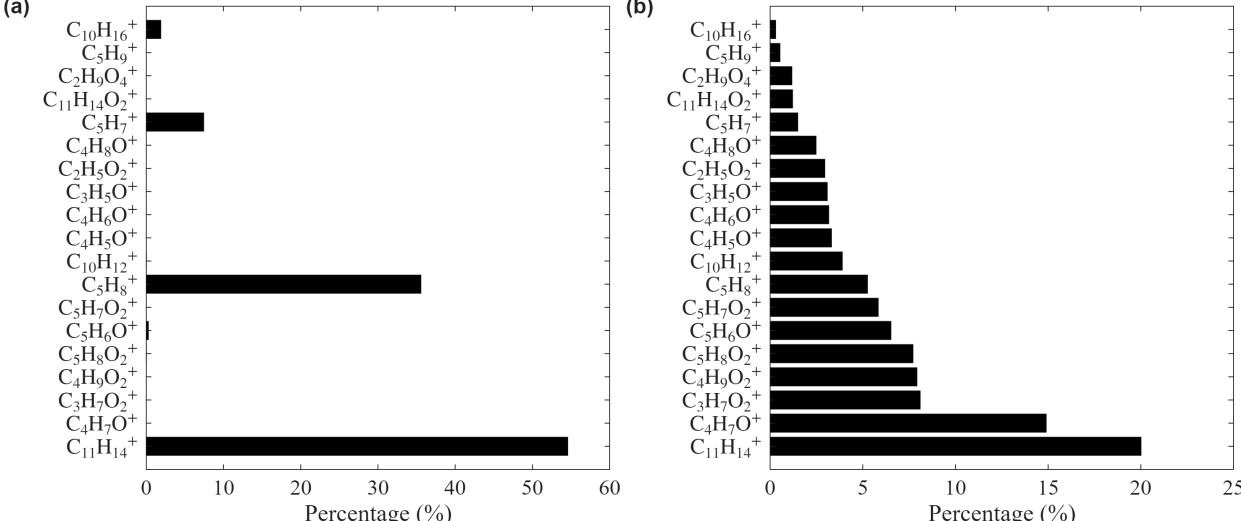

**Figure 2.** Distribution of isoprene product ions in different dilution gases (a) UHP $N_2$ and (b) zero air.

and because the reaction enthalpies for hydride abstraction from MACR and $NH_3$ are endothermic (Lias et al., 1988; Bargholz

et al., 2013). The adduct formation becomes the favorable pathway. The reported $C_6H_6^+$ affinities for NO and $NH_3$ are $184\pm21$

kJ mol$^{-1}$ and 78.7 kJ mol$^{-1}$, respectively (Reents and Freiser, 1980; Mizuse et al., 2010), both exceeding that of benzene.

This is consistent with our observation that adduct ion is the sole product ion for NO and $NH_3$. While the $C_6H_6^+$ affinity for

MACR is not available, the observed formation of the adduct ion suggests that its affinity is also greater than that of benzene.

Another analyte in the high IE class that we characterize is 2-Methyl-3-buten-2-ol (232 MBO, $C_5H_{10}O$). This compound is of

interest because it can interfere with isoprene measurements in other techniques, such as PTR-MS (Karl et al., 2012). Since the

thermodynamic properties of 232 MBO are largely unknown, we infer its ionization pathways based on the observed product

ions. The dominant product ion of 232 MBO is $C_{11}H_{14}^+$, which accounts for 80% of the total product ions. This observation

is consistent with a previous study by Leibrock and Huey (2000), which attributed the formation of $C_{11}H_{14}^+$ to 232 MBO

dehydration followed by adduct formation with $C_6H_6^+$ (i.e., $C_5H_{10}O - H_2O + C_6H_6^+$). Because this same ion ($C_{11}H_{14}^+$) is also

produced via the adduct ion formation between isoprene and $C_6H_6^+$, the presence of 232 MBO can interfere with isoprene

quantification using $C_{11}H_{14}^+$ in benzene CIMS. In addition to $C_{11}H_{14}^+$, we also detect the ion $C_5H_9O^+$, which is likely formed

through a hydride abstraction reaction. Adduct ion is not observed, which is consistent with its calculated $C_6H_6^+$ affinity (73.8

kJ mol$^{-1}$) slightly lower than that of benzene (Vermeuel, 2021).

We also tested six other compounds in the high IE class, including hydrogen cyanide, acetaldehyde, acetonitrile, methyl ethyl

ketone, acetone, and acrolein. The benzene CIMS exhibits negligible sensitivity toward these compounds. Similar behavior has

been observed for methanol, ethanol, and 2-propanol, which also fall into the high IE class (Leibrock and Huey, 2000). This

lack of sensitivity can be attributed to their IE exceeding 9.24 eV and their PA being lower than that of the phenyl radical (884

kJ mol$^{-1}$), rendering both charge transfer and proton transfer thermodynamically unfavorable. Although the $C_6H_6^+$ affinities of





**Table 1.** Absolute sensitivity, product ion fraction ($f_p$), normalized sensitivity ($\alpha$, $\beta$), background signal[a], and detection limits (LoD, 1 second and 1 minute integration times) for the analytes calibrated in this study.

| Class | Species | Product Ion | Ion $m/z$ | Sensitivity cps pptv$^{-1}$ | $f_p$ | $\alpha/10^3$ ncps pptv$^{-1}$ | $\beta/10^3$ ncps pptv$^{-1}$ | Back-ground (cps) | LoD pptv (1 sec) | LoD pptv (1 min) |
|---|---|---|---|---|---|---|---|---|---|---|
| low IE | 2-Methylfuran | $C_5H_6O^+$ | 82.0 | 6.9 | 1 | 11.3 | 2.4 | 14.1 | 2.0 | 0.3 |
| | 1,3,5-TMB | $C_9H_{12}^+$ | 120.1 | 11.1 | 1 | 14.3 | 3.4 | 4.9 | 0.6 | 0.1 |
| | o-Cresol | $C_7H_8O^+$ | 108.1 | 11.2 | 1 | 16.5 | 3.3 | 19.2 | 1.6 | 0.2 |
| | Limonene | $C_{10}H_{16}^+$ | 136.1 | 6.7 | 0.62 | 8.8 | 1.2 | 3.4 | 0.9 | 0.1 |
| | m-Xylene | $C_8H_{10}^+$ | 106.1 | 9.2 | 0.99 | 12.9 | 2.9 | 14.8 | 1.4 | 0.2 |
| | o-Xylene | $C_8H_{10}^+$ | 106.1 | 10.4 | 0.99 | 13.2 | 2.3 | 14.8 | 1.3 | 0.2 |
| mid IE | Toluene | $C_7H_8^+$ | 92.1 | 9.0 | 1 | 11.6 | 0.9 | 28.7 | 2.3 | 0.3 |
| | Chlorobenzene | $C_6H_5Cl^+$ | 112.0 | 7.8 | 1 | 11.7 | 0.0 | 0.4 | 0.4 | 0.1 |
| | D5-siloxane | $C_9H_{27}O_5Si_5^+$ | 355.1 | 7.9 | 1 | 11.1 | 0.0 | 0.9 | 0.5 | $7\times10^{-5}$ |
| | Dimethyl sulfide | $C_2H_6S^+$ | 62.0 | 4.3 | 0.52 | 5.6 | 0.0 | 4.7 | 3.7 | 0.5 |
| | | $C_8H_{12}S^+$ | 140.1 | 3.1 | 0.48 | 0.26 | 21.3 | 9.4 | 4.4 | 0.6 |
| | Isoprene | $C_4H_7O^+$ | 71.0 | 0.7 | 0.15 | 1.0 | 0.0 | 14.1 | 24.1 | 3.2 |
| | | $C_{11}H_{14}^+$ | 146.1 | 0.9 | 0.20 | 0.1 | 5.4 | 6.7 | 8.4 | 1.4 |
| high IE | NO | $C_6H_6NO^+$ | 108.0 | 3.1 | 1 | 2.9 | 0.0 | 62.2 | 26.9 | 5.0 |
| | Ammonia | $C_6H_9N^+$ | 95.1 | 1.6 | 1 | 2.0 | 2.1 | 208.3 | 85.2 | 67.4 |
| | MACR | $C_{10}H_{12}O^+$ | 148.1 | 3.3 | 1 | 5.2 | 0.0 | 15.4 | 5.0 | 0.9 |

[a] The calculation of background signal and detection limit is described in the SI Section 2.2.

these compounds are not reported, the absence of adduct ion formation suggests that their affinities are likely lower than that
of benzene.

All analytes investigated in previous studies are summarized in Figure S1, and their behavior aligns with the framework proposed here. Overall, categorizing analytes into three classes based on their IEs provides a useful framework for understanding the underlying ion chemistry and enhancing the interpretability of benzene CIMS measurements. As discussed later, analytes within the same class tend to exhibit similar sensitivity.

## 3 Instrument Performance

### 3.1 Absolute sensitivity

We calibrate the instrument sensitivities, expressed in counts per second per pptv (cps pptv$^{-1}$), for NO, NH$_3$, and twelve VOCs.
NO is introduced using a gas cylinder (3.01 ppmv$\pm$2%, Scott Air Gas LLC, USA), while the VOCs are introduced from two





custom gas mixtures provided by Apel-Riemer Environmental, Inc. (Table S4). For $NH_3$, we use a $NH_3$ wafer device with a
permeation rate of 82±21 ng min$^{-1}$ at 40°C (PDWF-0140, VICI Metronics Inc.). The calibration gases from the cylinders
and the permeation tube are diluted with dry zero air from a zero air generator (Tisch Environmental, Model ZA-747-30).
All calibrations were performed under the instrument's optimal operating conditions, as described in SI Section 2.4 where we
investigate the effects of IMR temperature, pressure, and flow rate of reagent gas on analyte sensitivities. The sensitivities of
the 14 analytes are listed in Table 1.

In the low IE class, both $C_6H_6^+$ and $(C_6H_6)_2^+$ serve as reagent ions, with charge transfer being the dominant ionization pathway
and the molecular ion $(X^+)$ as the primary product ion. Four aromatic compounds in this class (1,3,5-trimethylbenzene (1,3,5-
TMB), o-cresol, m-xylene, and o-xylene) exhibit similarly high sensitivities, ranging from 9.2 to 11.2 cps pptv$^{-1}$. These values
likely represent the upper limit of sensitivity near the collision limit. Limonene shows a slightly lower sensitivity of 6.7 cps
pptv$^{-1}$, due to fragmentation caused by dissociative charge transfer. The molecular ion constitutes 62% of the total product ion
signal. Accounting for this fragmentation, the corrected sensitivity is 6.7/0.62 = 10.8 cps pptv$^{-1}$, consistent with the sensitivity
range observed for other low IE analytes. 2-Methylfuran exhibits a sensitivity of 6.9 cps pptv$^{-1}$, which is lower than expected
for collision-limit ionization, but the reason is unclear.

In the mid IE class, analytes undergo charge transfer with $C_6H_6^+$ to form molecular ions and may also react with $(C_6H_6)_2^+$ to
produce adduct ions. For toluene, chlorobenzene, and D5-siloxane, analytes for which only charge transfer products are ob-
served, sensitivities range from 7.8 to 9.0 cps pptv$^{-1}$. These values are slightly lower than those observed for low IE analytes,
likely because only $C_6H_6^+$ participates in charge transfer for analytes in mid IE class, whereas both $C_6H_6^+$ and $(C_6H_6)_2^+$ con-
tribute in the low IE class. The DMS sensitivity, quantified using the charge transfer product $(C_2H_6S^+)$, is 4.3 cps pptv$^{-1}$, lower
than that of other mid IE analytes. This reduction may be attributed to mass-dependent transmission efficiency, as the $m/z$ of
$C_2H_6S^+$ is only 62. As estimated in SI Section 1.4, the transmission efficiency at $m/z$ 62 is 0.42, yielding a corrected sensitivity
of 10.1 cps pptv$^{-1}$, comparable to the others. The sensitivity of the DMS adduct ion is 3.1 cps pptv$^{-1}$, which will be discussed
alongside other adduct ions in high IE class. For isoprene, the sensitivity based on any single product ion is low due to its
complex product ion distribution (Figure 2).

In the high IE class, analytes cannot undergo charge transfer reactions; instead, other ionization pathways may occur. For
the three analytes we calibrate, NO, MACR, and $NH_3$, only adduct ions are observed. The sensitivities of NO and MACR are
3.1 and 3.3 cps pptv$^{-1}$, respectively, similar to that of DMS adduct ion (3.1 cps pptv$^{-1}$). $NH_3$ exhibits a lower sensitivity of
1.6 cps pptv$^{-1}$. This reduced sensitivity may arise from its lower $C_6H_6^+$ affinity (78.7 kJ mol$^{-1}$), compared with those of NO
and DMS (184 and 113.1 kJ mol$^{-1}$, respectively).

## 3.2 Normalized sensitivity

To account for variations in sensitivity due to fluctuations in ion source intensity, normalized sensitivity is typically used. This
is calculated by normalizing the signal of the product ion to that of the reagent ion(s). In previous benzene CIMS studies,
normalization has been performed using $C_6H_6^+$, $(C_6H_6)_2^+$, or the sum of both reagent ions (Kim et al., 2016; Lavi et al., 2018;
Riva et al., 2024; Aggarwal et al., 2025). As discussed in Section 2.2, the relevant reagent ion(s) vary with the analyte and





the ionization mechanism. Therefore, the choice of reagent ion(s) for normalization should be based on the analyte and its associated ionization pathway(s). Below, we derive a normalization method based on the example reactions shown in reactions

R10 and R11, where analyte X may react with both reagent ions to form a common product ion, $P^+$.

$$C_6H_6^+ + X \xrightarrow{k_1} f_1 P^+ + \text{others} \tag{R10}$$

$$(C_6H_6)_2^+ + X \xrightarrow{k_2} f_2 P^+ + \text{others} \tag{R11}$$

Assuming that the concentrations of both analytes and reagent ions remain constant during the ionization process, the number

concentration of $P^+$ produced from both reactions is given by

$$\Delta[P^+] = f_1 k_1 [C_6H_6^+][X]\Delta t + f_2 k_2 [(C_6H_6)_2^+][X]\Delta t \tag{1}$$

Here, [] represents the number concentration in the IMR, $f_1$ and $f_2$ represent the yield of product ion for each reaction, $k_1$ and $k_2$ represent the corresponding rate constants, and $\Delta t$ is the reaction time. To relate the ion concentration in the IMR to the measured ion count rate I (in cps), we introduce an efficiency factor E, which accounts for transmission through the ion optics

and the detection efficiency of the electron multiplier. E is assumed constant for each ion under fixed instrument conditions. We also express the analyte concentration [X] as the product of its volume mixing ratio $C_X$ (in pptv) and the number density of air in the IMR N. Substituting into Equation 1 gives:

$$\frac{I_{P^+}}{E_{P^+}} = f_1 k_1 \frac{I_{C_6H_6^+}}{E_{C_6H_6^+}} C_X N \Delta t + f_2 k_2 \frac{I_{(C_6H_6)_2^+}}{E_{(C_6H_6)_2^+}} C_X N \Delta t \tag{2}$$

Next, we consolidate constants into two parameters, $\alpha$ and $\beta$, defined as $\alpha = f_1 k_1 N \Delta t \frac{E_{P^+}}{E_{C_6H_6^+}}$ and $\beta = f_2 k_2 N \Delta t \frac{E_{P^+}}{E_{(C_6H_6)_2^+}}$.

Inserting these into Equation 2 and introducing a scaling factor of $10^6$ for practical units (i.e., counts per million reagent ion counts), we obtain:

$$\frac{I_{P^+}}{C_X} = \alpha \frac{I_{C_6H_6^+}}{10^6} + \beta \frac{I_{(C_6H_6)_2^+}}{10^6} \tag{3}$$

We can further rearrange Equation 3 to isolate the normalized sensitivity. When $C_6H_6^+$ is the primary reagent ion, Equation 3 can be expressed as

$$\frac{\frac{I_{P^+}}{\frac{I_{C_6H_6^+}}{10^6} + \frac{\beta}{\alpha} \frac{I_{(C_6H_6)_2^+}}{10^6}}}{C_X} = \alpha \tag{4}$$



Here, $\alpha$ is the normalized sensitivity (in ncps pptv$^{-1}$), with the denominator consisting of a weighted sum of the reagent ion signals. Conversely, when $(C_6H_6)_2^+$ is the dominant reagent ion, the expression becomes

$$\frac{\frac{I_{P_1^+}}{\frac{\alpha}{\beta}\frac{I_{C_6H_6^+}}{10^6} + \frac{I_{(C_6H_6)_2^+}}{10^6}}}{C_X} = \beta \tag{5}$$

To determine $\alpha$ and $\beta$, we conduct calibration experiments in which the analyte mixing ratio ($C_X$) is held constant while
the temperature of the benzene permeation tube is varied. This adjustment alters the benzene concentration in the IMR and
consequently the relative abundances of the reagent ions. We then examine the response of the product ion signal ($I_{P^+}$) to
changes in the reagent ion signals ($I_{C_6H_6^+}$ and $I_{(C_6H_6)_2^+}$) by performing a multivariate linear regression based on Equation 3. The
fitted values of $\alpha$ and $\beta$ for all calibrated analytes are presented in Table 1, with representative regression fits shown in Figures
S5 - S7.

While the $\alpha$ and $\beta$ values represent normalized sensitivities, they also provide insights into the underlying ion chemistry. For
analytes in the low-IE class, $\alpha$ values range from 12.9 to 16.5 (after correcting for the product ion distribution of limonene),
which are consistently larger than the corresponding $\beta$ values (1.2–3.4). This likely reflects the faster charge transfer reaction
with $C_6H_6^+$ compared to $(C_6H_6)_2^+$. For toluene, chlorobenzene, and D5-siloxane, three analytes in the mid-IE class, their $\beta$
values are either zero or significantly lower than the corresponding $\alpha$ values. This is consistent with negligible charge transfer
between these analytes and $(C_6H_6)_2^+$. The rate of charge transfer is related to the difference in IE between the analyte and the
reagent ion. Stone and Lin (1980) shows that the charge transfer rate with $C_6H_6^+$ is fast and plateaus when the analyte's IE
is much lower than that of benzene, but decreases rapidly as this difference narrows. A similar trend is observed for charge
transfer ions in this study as shown in Figure S4. Lastly, for analytes that undergo adduct formation, no clear trend emerges in
the $\alpha$ and $\beta$ values. For DMS, NO and MACR, their $\alpha$ values are larger than $\beta$ values, which are zero. In contrast, for NH$_3$, $\alpha$
and $\beta$ are similar, both of which are non-zero. The underlying reasons for these differing behaviors among the four analytes
remain unclear.

### 3.3  Humidity dependence

Previous studies (Kim et al., 2016; Lavi et al., 2018) have shown that water has complex effects on reagent ion distribution
and analyte sensitivity. They also observed protonated water clusters, although their formation mechanisms remain unclear. To
investigate this, we sample UHP N$_2$ at varying humidity levels. Under dry conditions, only $C_6H_6^+$ and $(C_6H_6)_2^+$ are detected.
As humidity increases, we observe an exponential increase in the signals of protonated water clusters $(H_2O)_nH^+$ for n = 3 and
4, with negligible signals for n = 1, 2, or $\geq$ 5 (Figure S10). No benzene–water cluster ions (i.e., $C_6H_6(H_2O)_n^+$) are observed.
The formation and detection of these ions results from the combined influences of benzene–water ion chemistry, declustering
effects, and mass-dependent transmission efficiency. Figure 3 shows a scheme to illustrate the proposed formation mechanism
of protonated water clusters and the thermodynamic properties of water clusters are summarized in Tables S6 and S7.



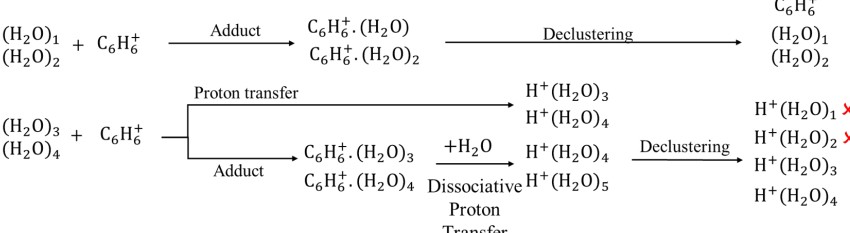

**Figure 3.** Schematic illustration of the formation mechanisms of protonated water clusters. The mechanisms include (1) proton transfer between $C_6H_6^+$ and water trimers or tetramers and (2) dissociative proton transfer between $C_6H_6(H_2O)_n^+$ (n = 3 and 4) and water.

Water vapor can form clusters in the atmosphere. Assuming a water mixing ratio of 2% (corresponding to 100% relative humidity at 18 °C), the estimated mixing ratios of the clusters are as follows: monomer at 20,000 ppmv, dimer at 20 ppmv, trimer at 0.2 ppmv, tetramer at 0.02 ppmv (Dunn et al., 2004). Pentamer and larger clusters are not considered because of their low abundances. Based on the calibration experiment described in Section 3.2, we find that $C_6H_6^+$ is the primary reagent responsible for generating protonated water clusters (Figure S11). Thus, the following discussion focuses on the reactions between $C_6H_6^+$ and $(H_2O)_n$ (n = 1-4).

$(H_2O)_n$ (n = 1-4) cannot undergo charge transfer with $C_6H_6^+$, as their IEs exceeds that of benzene (Table S6). The hydride abstraction is also unlikely. Proton transfer and adduct formation are possible for some water clusters. Based on their thermodynamic properties, we find that water monomer and dimer share similar ionization pathways, while water trimer and tetramer follow different pathways. Water trimer and tetramer have PA values comparable to or exceeding that of the phenyl radical, leading to the formation of $(H_2O)_3H^+$ and $(H_2O)_4H^+$. In addition, the $C_6H_6^+$ affinities of water trimer and tetramer are comparable to or exceeding that of benzene, leading to the formation of adduct ions $C_6H_6(H_2O)_3^+$ and $C_6H_6(H_2O)_4^+$. Regarding water monomer and dimer, they are unlikely to undergo proton transfer with $C_6H_6^+$ because their PA values are lower than that of the phenyl radical. Their $C_6H_6^+$ affinities are lower than benzene, but adduct ions $C_6H_6(H_2O)^+$ and $C_6H_6(H_2O)_2^+$ may still form under humid conditions, where the chemistry of water monomer and dimer overruns that of benzene in the IMR. In summary, product ions $C_6H_6(H_2O)_n^+$ (n = 1–4) and $(H_2O)_nH^+$ (n = 3 and 4) can form in the IMR.

These product ions can undergo further reactions in the IMR. The benzene–water adduct ions $C_6H_6(H_2O)_n^+$ (n = 3 and 4) may collide with water molecules and undergo a dissociative proton transfer reaction (Ibrahim et al., 2005), resulting in the formation of $(H_2O)_nH^+$ (n = 4 and 5). This reaction is thermodynamically unfavorable for n = 1 and 2 (Table S6). Consequently, the initially formed adduct ions are converted into $C_6H_6(H_2O)_n^+$ (n = 1 and 2) and $(H_2O)_nH^+$ (n = 3–5).

These ions may further decluster in the ion optics. The extent of declustering depends on their binding energy, but a quantitative relationship remains unknown. We qualitatively infer the extent of declustering from the voltage scanning test (SI Section 2.6). Figure S14b shows that the signals of $(C_6H_6)_2^+$, $H_9O_4^+$, and $H_7O_3^+$ decrease by roughly 80% when the voltage gradient between the BSQ front and the skimmer is increased by 5 V. These three ions have binding energies of 74, 73, and 85 kJ mol$^{-1}$, respectively. Based on this, we infer that ions with binding energies below 70 kJ mol$^{-1}$ likely undergo extensive declustering





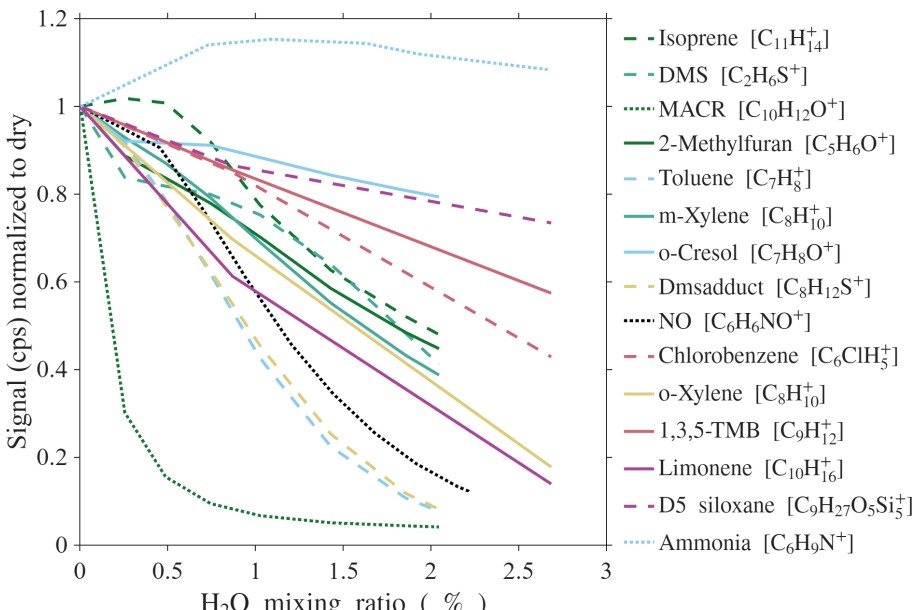

**Figure 4.** Humidity-dependent sensitivity of analytes. Solid, dashed, and dotted lines represent analytes in low, mid, and high IE classes, respectively.

in our instrument. The adduct ions $C_6H_6(H_2O)_n^+$ (n = 1 and 2) have binding energies of 38 and 59 kJ mol$^{-1}$, respectively, indicating that they decluster in the instrument, which likely explains why they are not observed. The binding energies of $(H_2O)_nH^+$ are 84, 73, and 56 kJ mol$^{-1}$ for n = 3, 4, and 5, respectively. Thus, $(H_2O)_5H^+$ likely undergoes extensive declustering, while partial declustering is expected for n = 3 and 4. This process may yield $(H_2O)_nH^+$ with n = 1 and 2. However, these two ions

345 are not observed, likely due to poor transmission efficiency at low *m/z* in the instrument (Figure S2c).

Overall, the protonated water clusters $(H_2O)_nH^+$ (n = 3 and 4) are likely formed via two pathways: proton transfer from $C_6H_6^+$ to water trimer and tetramer, and dissociative proton transfer from the adduct ions $C_6H_6(H_2O)_n^+$ (n = 3 and 4). The absence of benzene–water cluster ions is attributed to their weak binding energies, leading to declustering within the instrument. The lack of detectable $(H_2O)_nH^+$ ions with n=1 and 2 is likely due to low transmission efficiency at low *m/z*.

350 We investigate the dependence of analyte sensitivity on sample humidity by diluting the calibration gas with humidified air. The humidified air is generated by flowing zero air through a frit bubbler containing HPLC-grade water. The water mixing ratio of sample is monitored using a Picarro G2401 gas concentration analyzer (SI Section 3.2). The maximum water mixing ratio of sample reached is 2.75%, equivalent to 100% relative humidity at 23 °C. The dependence of analyte sensitivity on sample absolute humidity is shown in Figure 4. The majority of analytes exhibit a decreasing sensitivity with increasing humidity, with

355 $NH_3$ being an exception.

One notable finding is that water ion signals in CIMS are influenced not only by sample humidity but also by the analyte mixing ratio. In one experiment, we hold the water mixing ratio constant at 1.25% and vary the NO mixing ratio from 3 to





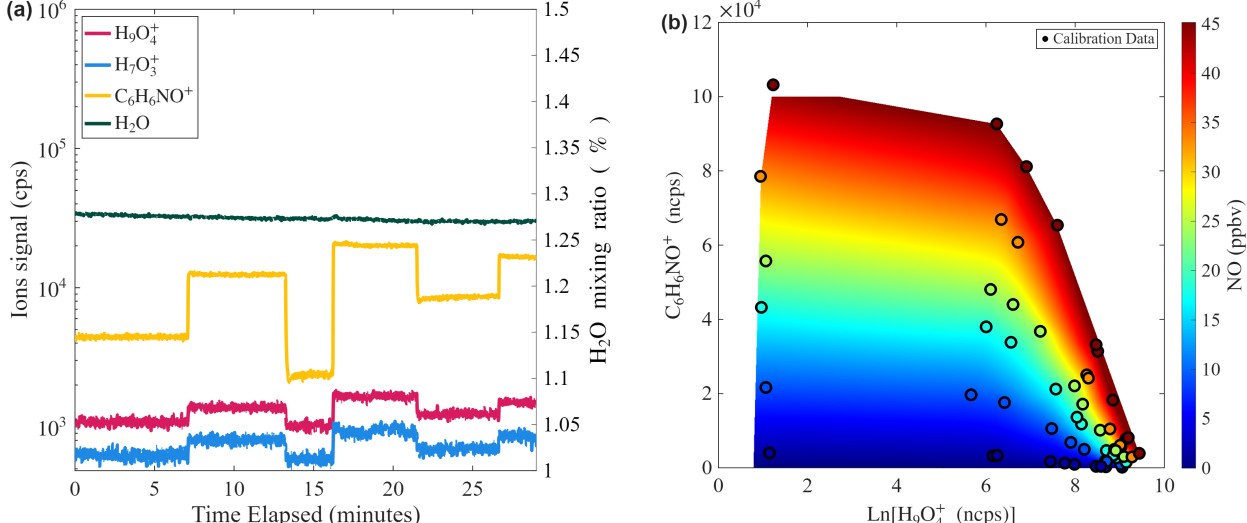

**Figure 5.** (a) Time series of water mixing ratio and signals of NO ($C_6H_6NO^+$) and water (e.g., $H_7O_3^+$ and $H_9O_4^+$) during a calibration experiment where water mixing ratio is held constant while NO mixing ratio is varied from 3 to 15 ppbv. The water mixing ratio is measured by a PICARRO G2401 gas concentration analyzer. (b) Two-dimensional calibration contour interpolated from calibration points obtained in the humidity-dependent calibration of NO.

15 ppbv. Despite constant water mixing ratio, the $H_9O_4^+$ signal changes with NO mixing ratio. A similar effect is observed for VOCs (Figure S12a) and was also reported by Lavi et al. (2018). The reason for this phenomenon is unclear, but may be
explained by clustering between $C_6H_6^+$, analyte, and water to form weakly bound adducts $C_6H_6X(H_2O)_n^+$, which subsequently undergo declustering, producing $(H_2O)_nH^+$ ions. As a result, higher analyte mixing ratio leads to higher water ion signals.

To correct for humidity-dependent sensitivity, the ideal approach is to directly monitor the water mixing ratio. However, in cases where it is not measured, water ion signals measured by CIMS can be used as a proxy, which is a widely adopted method in CIMS. However, this approach becomes challenging when water ion signals are influenced not only by sample humidity but
also by the analyte mixing ratio. To address this, we develop an interpolation-based correction method. This method begins with calibration experiments covering a wide range of mixing ratios of analyte and water. These conditions are obtained through two sets of experiments: in the first, water mixing ratio is varied while the analyte mixing ratio is held constant; in the second, the analyte mixing ratio is varied while water mixing ratio is kept constant. Using the resulting calibration data, we plot the analyte product ion signal as a function of the water ion signal, color-coded by the analyte mixing ratio. The dataset is then interpolated
into a two-dimensional surface, referred to as the calibration contour, using MATLAB's natural neighbor interpolation method (Figure 5b). Finally, this calibration contour allows us to infer the analyte mixing ratio in ambient measurements based on observed signals of analyte product ion and water ion. The performance of this correction method will be evaluated using ambient data as discussed in Section 4.





# 4  Ambient measurements

We deployed the benzene CIMS instrument for one week each in Chicago, IL (July 2024) and St. Louis, MO (August 2024) to evaluate its performance. The Chicago site is located on the University of Illinois Chicago campus, approximately 0.3 km from Interstate 90 and 2.5 km from downtown Chicago. The St. Louis site is located on the Washington University in St. Louis campus, adjacent to a major roadway (Skinker Boulevard, 2 lanes in each direction) and a large forested park (Forest park). These two locations offer diverse testing environments for the instrument. We compared the isoprene and NO measured by

CIMS to co-located reference measurements. NO was measured with a Thermo Scientific Model 42i $NO-NO_2-NO_x$ analyzer at a 1 minute time resolution, while isoprene was measured with a gas chromatograph with photoionization detection (GC-PID, SRI model 8610C) at a 25-minute resolution (SI Section 3.3).

## 4.1  Measurements of NO

Figures 6a and 7a show the time series of NO mixing ratios measured by both the benzene CIMS and the $NO_x$ analyzer at the

St. Louis and Chicago sites, respectively. The average NO mixing ratio was higher in Chicago (2.1 ppbv) than in St. Louis (0.6 ppbv), due to heavier traffic near the Chicago site. At both locations, NO exhibited a clear morning rush hour peak (Figures 6c and 7c). The benzene CIMS measurements agreed well with those from the $NO_x$ analyzer in St. Louis (Figure 6), with an $R^2$ value of 0.92 and a slope of 0.97. The comparison was slightly weaker in Chicago, where the scatter plot was more dispersed (Figure 7b), but the overall agreement remained strong, with an $R^2$ of 0.92 and a slope of 0.92. Data points are color-coded by

the natural logarithm of the $H_9O_4^+$ signal, a proxy for sample humidity. No systematic dependence of measurement differences on $Ln(H_9O_4^+)$ is observed, supporting the effectiveness of the interpolation-based humidity correction method described in Section 3.3.

The inset in Figure 6a zooms in on a period with low NO mixing ratio. It shows that the signal-to-noise ratio of the benzene CIMS clearly exceeds that of the commercial $NO_x$ analyzer. Table 2 compares the performance of the benzene CIMS with four

CL $NO_x$ analyzers (Thermo 42i, Teledyne API T200U, AECOM Serinus 40, and Ecophysics CLD 780 TR), one research-grade CL analyzer (Ridley and Grahek, 1990), and one LIF NO analyzer (Rollins et al., 2020). The LIF instrument demonstrates the best overall performance, with the highest precision, lowest zero noise, and lowest detection limit. The benzene CIMS, research-grade CL analyzer, and Ecophysics CLD 780 TR show comparable performance, with single-pptv-level detection limits. In contrast, the other three commercial $NO_x$ analyzers exhibit higher zero noise and detection limits, typically in the

tens to hundreds of pptv range.

## 4.2  Measurements of Isoprene

As shown in Figure 2, isoprene exhibits a complex distribution of product ions in the benzene CIMS. For quantification, we select the adduct ion $C_{11}H_{14}^+$, which represents the largest fraction (about 20%) of the total product ion signal. Several oxygenated product ions, including $C_4H_7O^+$ and $C_5H_8O_2^+$, contribute a notable portion of the overall signal. However, quantifying isoprene




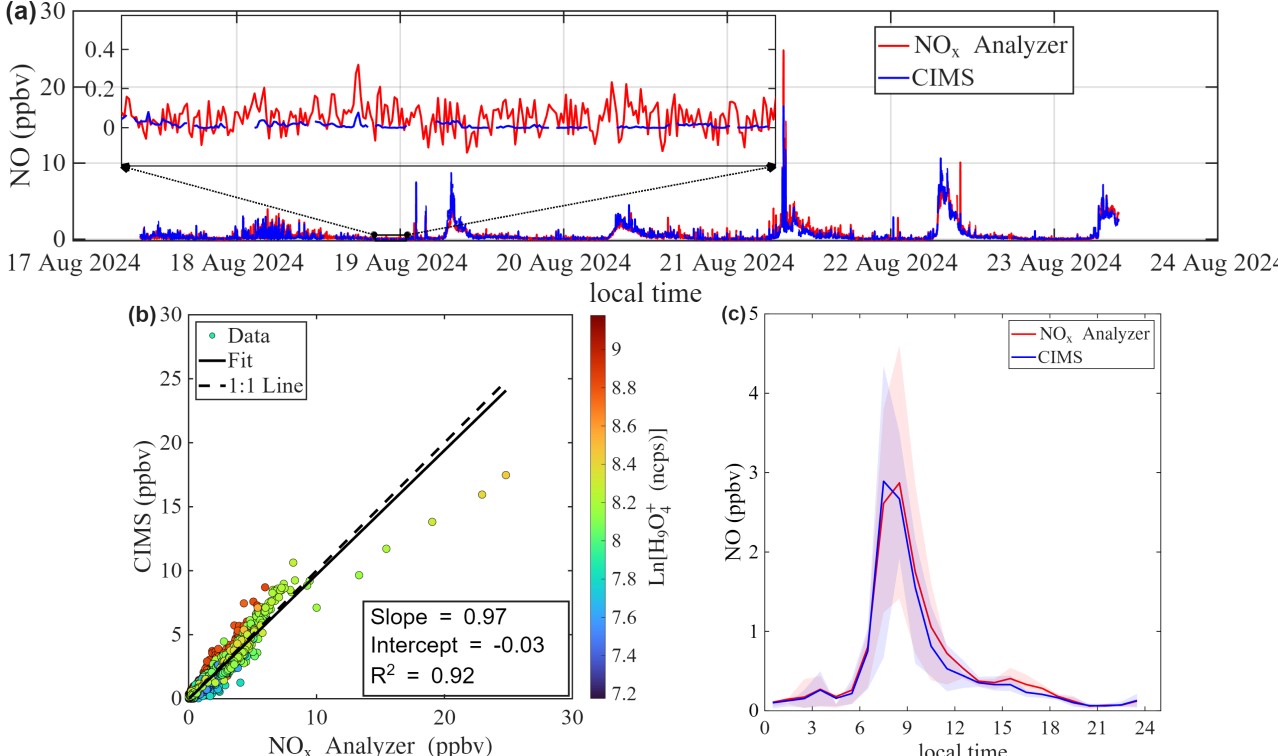

**Figure 6.** Intercomparison of NO mixing ratios measured by benzene CIMS and $NO_x$ analyzer. The 1 Hz CIMS data are averaged to 1-minute resolution to match that of the $NO_x$ analyzer. (a) Time series of NO during ambient sampling in St. Louis. (b) Scatter plot comparing NO measurements from the two instruments. Data points are colored by the natural log signal of the water tetramer ion. (c) Diurnal trend of NO mixing ratio represented by the median value. Shaded areas represent the 25th and 75th quartiles.

based on these oxygenated ions poses challenges due to potential interferences from oxygenated volatile organic compounds (OVOCs) present in ambient air.

Figure 8a shows the time series of isoprene mixing ratios in St. Louis. Overall, the benzene CIMS measurements agree well with those from the GC-PID, yielding a linear regression of $[isoprene]_{CIMS} = 0.93 \times [isoprene]_{GC\text{-}PID} + 0.18$ and a $R^2$ of 0.99. Closer inspection of the time series reveals that the CIMS-reported isoprene was slightly lower than the GC-PID on the first day of measurements but became slightly higher during the latter part of the campaign. This shift may contribute to the slope deviating from unity and the non-zero intercept. This shift could be caused by drift in instrument sensitivity or potential interferences in the CIMS measurement. However, calibrations were not performed frequently enough to evaluate the stability of CIMS and GC-PID sensitivities. The diurnal trend of isoprene (Figure 8c) accentuates the period when CIMS reported higher values.

The agreement between the CIMS and GC-PID measurements is worse in Chicago compared to St. Louis. As shown in Figure 9a, isoprene mixing ratios measured by CIMS are generally higher than those measured by the GC-PID, and the



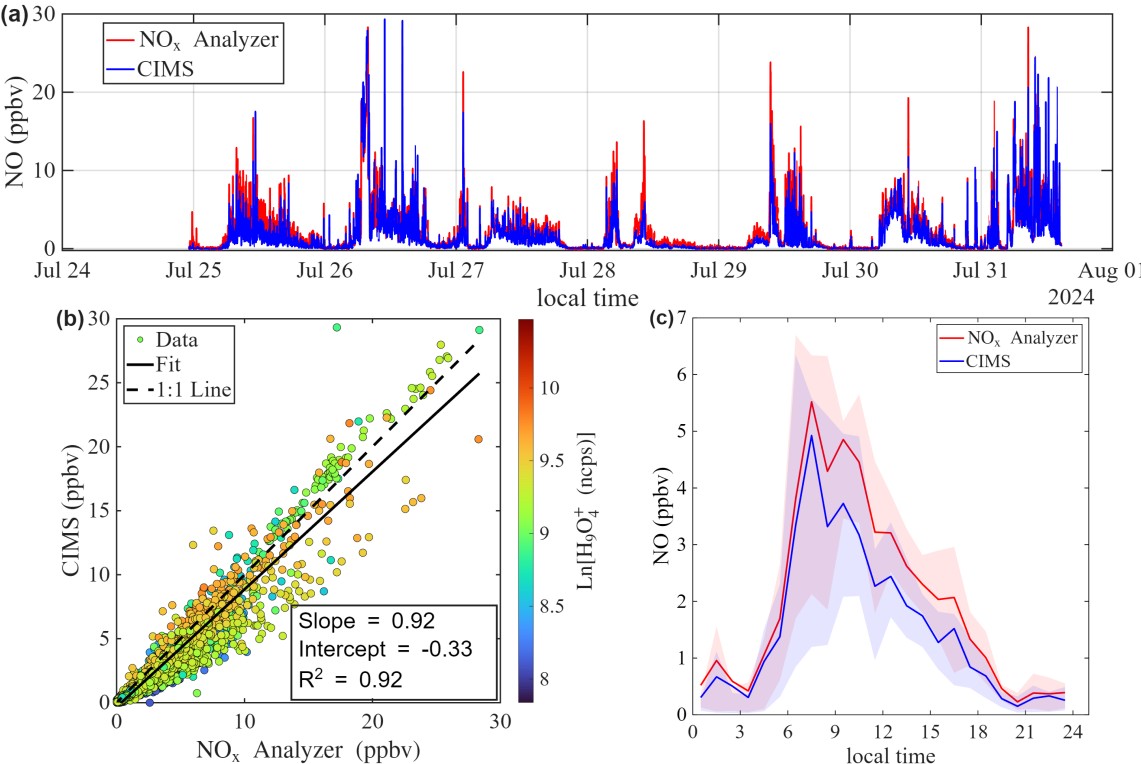

**Figure 7.** Same as Figure 6, but for the Chicago site.

**Table 2.** Specifications of the benzene CIMS, CL, and LIF instruments used for NO measurement. Information for the LIF and the research grade CL instruments are from Rollins et al. (2020) and Ridley and Grahek (1990), respectively. Information for commercial instruments is sourced from product brochures. The precision, zero noise, and detection limits correspond to 1 minute integration time. Precision is calculated using normalized adjacent differences, and detection limits are defined as 3 standard deviations ($\sigma$) of the background signal. Further details on the calculation of precision and detection limits are provided in SI Section 2.2.

|  | Benzene CIMS | NOAA LIF | Research grade CL | Eco Physics CLD 780 TR | Teledyne API Model T200U | ACOEM Serinus 40 | Thermo Model 42i |
|---|---|---|---|---|---|---|---|
| Precision (%) | 0.35 | 0.2 | - | - | 0.5 | 0.5 | 0.8 |
| Zero noise (pptv) | 0.8 | 0.08 | 1 | <25 | < 25 | <200 | 200 |
| Detection limit (pptv) | 5 | 0.24 | < 1 | 3 | < 50 | <400 | 400 |
| Power consumption (W) | 800 | 400 | - | 200 | 110 | 190 | 300 |
| Mass (Kg) | 250 | 50 | 90 -120 | 35 | 28 | 21.9 | 25 |

correlation between the two instruments is weak ($R^2 = 0.34$). This observation is consistent with findings from Leibrock et al. (2003), which reported that CIMS-measured isoprene levels in Boulder, CO were frequently higher than those measured by GC. That discrepancy was attributed to interferences from anthropogenic emissions from the Denver metropolitan area.




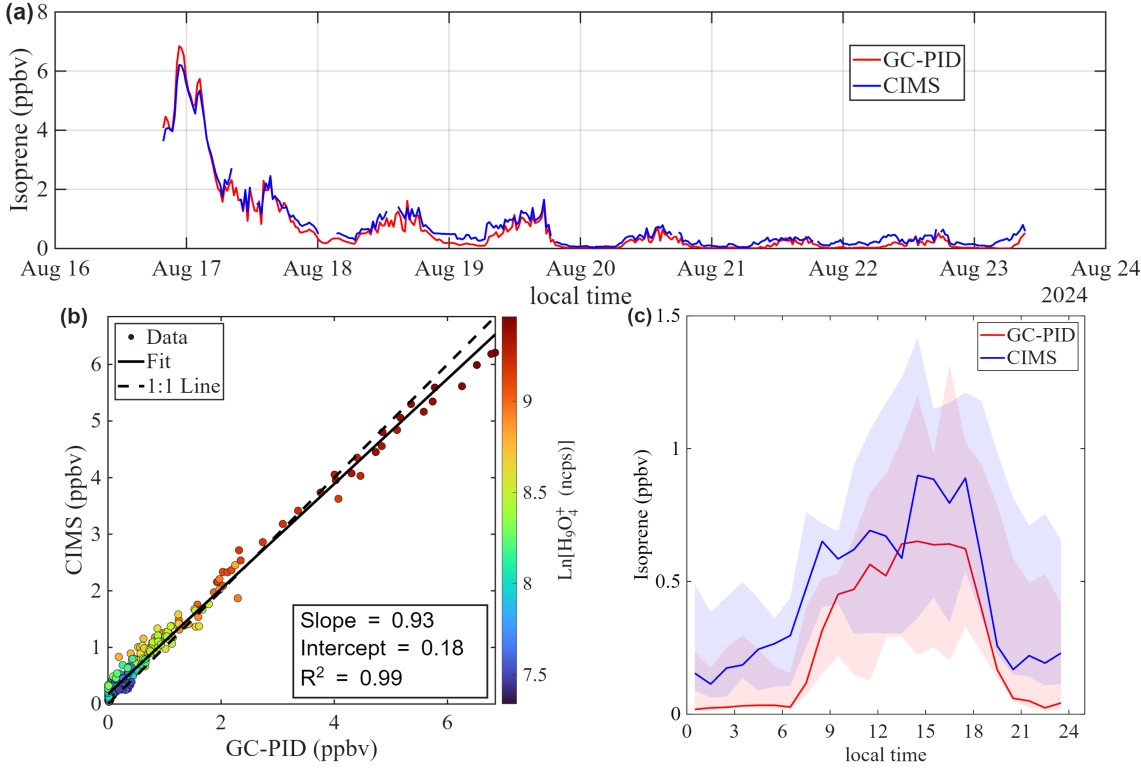

**Figure 8.** Intercomparison of Isoprene mixing ratios measured by benzene CIMS and GC-PID. The 1 Hz CIMS data are averaged to 10 minute sample collection period of GC-PID. (a) Time series of isoprene during ambient sampling in St. Louis. (b) Scatter plot comparing isoprene measurements from the two instruments. Data points are colored by the natural log signal of the water tetramer ion. (c) Diurnal trend of isoprene mixing ratio represented by the median value. Shaded areas represent the 25th and 75th quartiles.

A similar explanation likely applies here, as the Chicago site is located in a polluted urban area where CIMS measurements could be influenced by anthropogenic emissions such as 1,3-pentadiene, an isomer of isoprene. Taken together, the results from Leibrock et al. (2003) and this study highlight the challenges of using benzene CIMS to measure isoprene in polluted environments.

### 4.3 Measurements of other analytes

The benzene CIMS detected nearly 500 ions above the detection limit in both St. Louis and Chicago. Figure 10 presents a mass defect plot based on the St. Louis dataset, illustrating the instrument's measurement capabilities. In this plot, each ion is represented by a symbol denoting its chemical family, with color indicating its ionization pathway. Product ions are categorized as adduct, charge transfer, or proton transfer / hydride abstraction products based on their carbon number, double bond equivalence, and whether they contain an even or odd number of hydrogen atoms. Since we cannot distinguish between

proton transfer and hydride abstraction products, these two pathways are grouped together. The detailed classification procedure



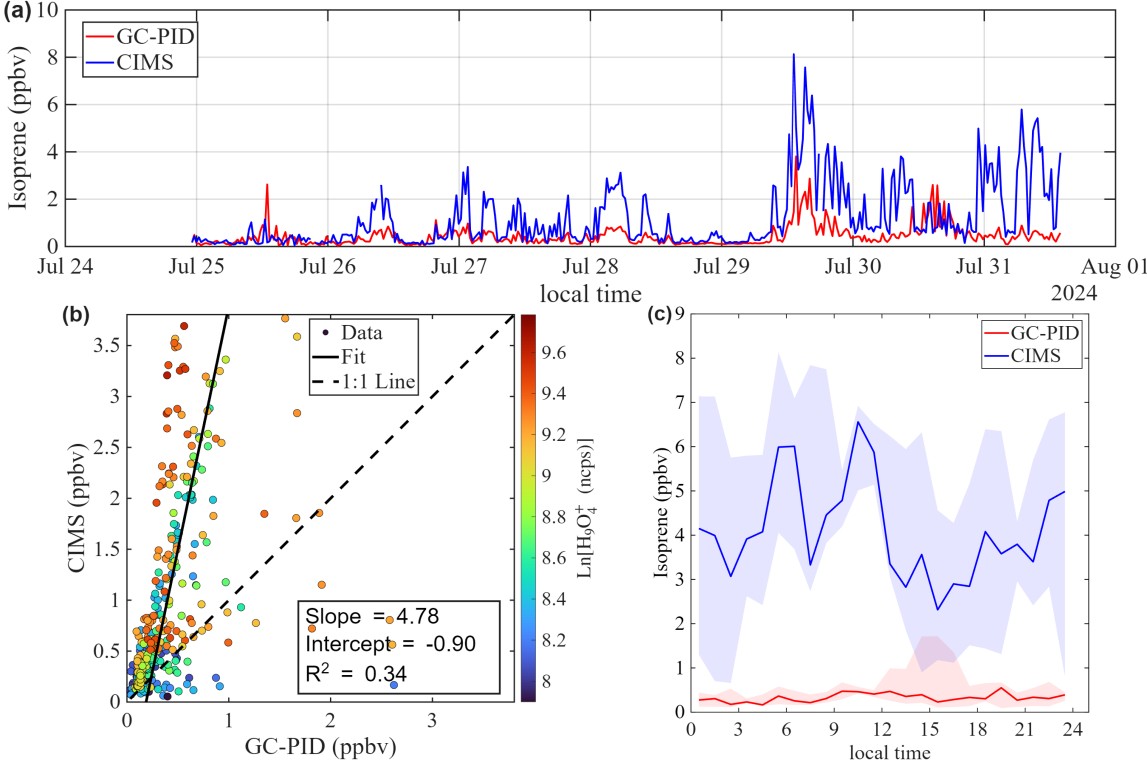

**Figure 9.** Same as Figure 8, but for the Chicago site.

is provided in SI Section 3.1. Note that a single analyte may contribute to multiple product ions. Among all detected ions, adduct ions account for the largest fraction (40%), followed by charge transfer (33%) and proton transfer/hydride abstraction ions (33%). Within the charge transfer category, 72% of the ions are in the form $C_xH_yO_z^+$ and 24% as $C_xH_y^+$. For adduct ions, 54% are $C_xH_yO_z^+$ and 27% are hydrocarbons ($C_xH_y^+$). 71% of the ions produced via proton transfer/hydride abstraction are of

the form $C_xH_yO_z^+$.

Within the oxygenated compounds, we observe ions corresponding to known isoprene oxidation products. These include $C_{10}H_{12}O^+$, likely representing adduct ions of MACR and methyl vinyl ketone (MVK), $C_5H_{10}O_3^+$ corresponding to charge transfer ions of isoprene epoxydiol (IEPOX) and isoprene hydroxy hydroperoxides (ISOPOOHs), and $C_5H_{10}O_4^+$ corresponding to charge transfer ions of isoprene hydroxycarbonyls. The time series and diurnal patterns of these ions are presented in

Figure 11. Further identification and speciation of these ions will require complementary approaches, including thermodynamic property calculations, GC for isomer separation, and laboratory isoprene oxidation experiments.





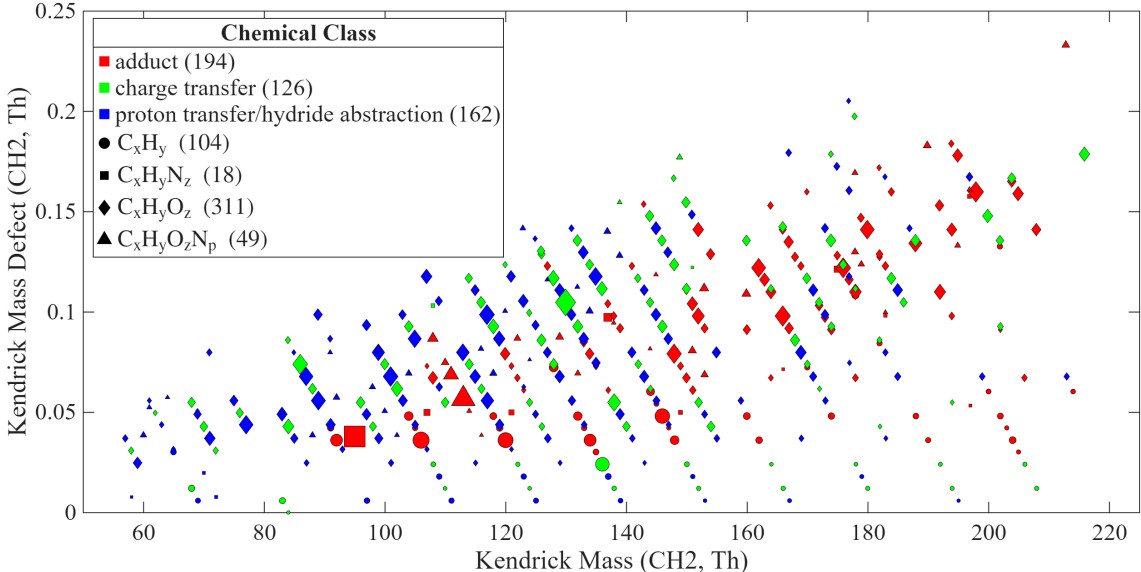

**Figure 10.** Mass defect plot using $CH_2$ as the reference unit, based on the St. Louis dataset. Each point represents an individual ion, with symbol shape denoting its chemical family and color indicating its ionization pathway. Products from proton transfer and hydride abstraction are grouped together, as they cannot be distinguished. The number of ions corresponding to each chemical family and ionization pathway is provided in parentheses in the legend. Symbol size is scaled to the square root of the campaign-averaged signal. A total of 482 ions with signals above the detection limit are shown.

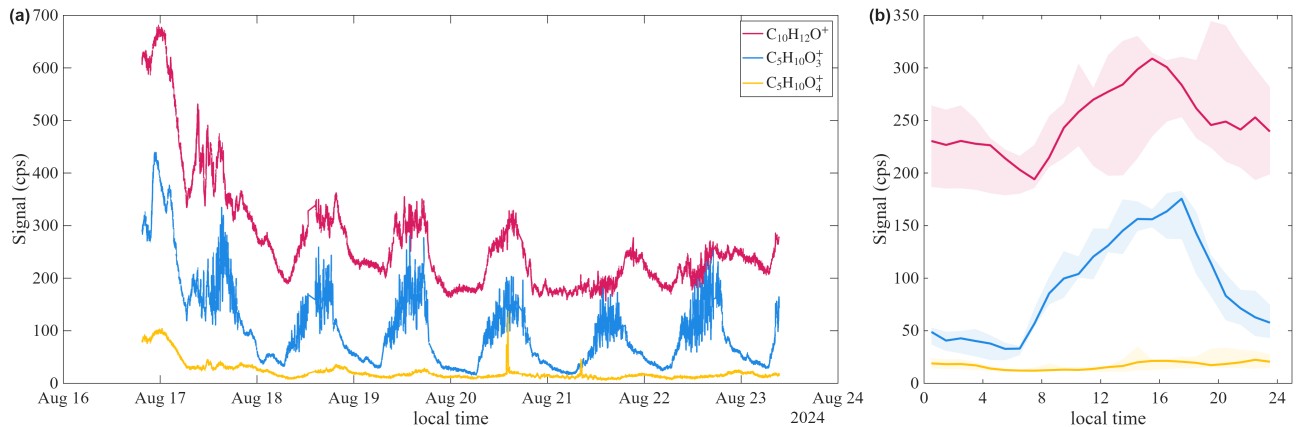

**Figure 11.** (a) Time series and (b) diurnal trends of potential isoprene oxidation products detected by benzene CIMS in St. Louis.



## 5 Conclusions

In this study, we characterize the performance of a CIMS using benzene cations as reagent ions through laboratory calibrations and ambient measurements. While previous studies have examined the performance of benzene CIMS for specific analytes, this work provides a systematic evaluation of its capability to detect a broad range of atmospheric species.

The ion chemistry of benzene cations is complex due to the presence of two reagent ions and multiple competing ionization pathways. Fundamentally, the competition between these ionization pathways is governed by key thermodynamic properties of the analytes, including ionization energy, proton affinity, $C_6H_6^+$ affinity, and hydride abstraction potential. To help interpret this complexity, we introduce a framework that enables qualitative prediction of an analyte's ion chemistry. In this framework, analytes are categorized into three classes based on their IEs. Analytes within the same class tend to exhibit similar ionization mechanisms and sensitivities.

For analytes in the low IE class, those with IE lower than those of both the benzene monomer and dimer, the dominant ionization pathway is charge transfer with both reagent ions, resulting in the molecular ion as the primary product ion. Fragmentation of the molecular ion may occur if the appearance energy of a fragment ion is lower than the ionization energy of the reagent ion. Analytes in this class exhibit the highest absolute sensitivities among all classes, typically ranging from 9.2 to 11.2 cps pptv$^{-1}$, which likely represents the collision-limit sensitivity. Apparent deviations from this range may arise due to reduced transmission efficiency or fragmentation of the molecular ion.

For analytes in the mid IE class, those with IE higher than that of the benzene dimer but lower than that of the benzene monomer, the ionization pathways depend on the reagent ion. These analytes can undergo charge transfer with $C_6H_6^+$ to form molecular ions, or they may undergo ligand exchange reactions with $(C_6H_6)_2^+$ to form adduct ions, if their $C_6H_6^+$ affinity exceeds that of benzene. Consequently, the product ion distribution is highly sensitive to the relative abundance of $C_6H_6^+$ and $(C_6H_6)_2^+$, which likely explains the variability in product ion distrbutions for the same analyte (e.g., isoprene and DMS) between this study and previous studies. For analytes where the molecular ion is the dominant product (e.g., toluene, D5-siloxane, and chlorobenzene), absolute sensitivities range from 7.8 to 9.0 cps pptv$^{-1}$. These values are slightly lower than those of low IE analytes, likely because only $C_6H_6^+$ contributes to charge transfer in the mid IE class, whereas both $C_6H_6^+$ and $(C_6H_6)_2^+$ contribute in the low IE class.

For analytes in the high IE class, those with IE greater than both the benzene monomer and dimer, charge transfer with either reagent ion does not occur. Instead, other ionization pathways such as proton transfer, adduct formation, and hydride abstraction may occur, depending on the thermodynamic properties of the analytes. Adduct formation is possible if the analyte has a higher $C_6H_6^+$ affinity than benzene. The absolute sensitivities for adduct ions fall within a narrow range (1.6–3.3 cps pptv$^{-1}$) for the four analytes examined here (DMS, NO, and MACR, and NH$_3$). Hydride abstraction may also occur (e.g., for 232-MBO), but the lack of relevant thermodynamic data limits further evaluation of this pathway.

The proposed framework provides a useful tool for diagnosing measurement capability, predicting ionization pathways and product ions, and constraining the sensitivity, provided their key thermodynamic properties are known. For example, benzene CIMS is unlikely to be sensitive to analytes with IE larger than benzene, PA smaller than phenyl radical, and $C_6H_6^+$ affinity



smaller than benzene. For analytes with IE smaller than benzene dimer (i.e., low IE class), charge transfer is likely the predominant ionization pathway. Their absolute sensitivities can be estimated at roughly $10\pm2$ cps pptv$^{-1}$, assuming minimal fragmentation and high transmission efficiency. This framework also informs strategies to optimize instrument sensitivity for specific analytes. For instance, a higher abundance of $C_6H_6^+$ should be targeted for analytes with $\alpha$ values greater than $\beta$, as is
typical for low IE species. If reducing fragmentation is a priority, the abundance of $(C_6H_6)_2^+$ in the IMR should be maximized, which has softer ionization chemistry than $C_6H_6^+$.

We acknowledge that the proposed framework has uncertainties and limitations that can be addressed in future work. First, calibrating a broader range of analytes will test the robustness of the framework and help refine uncertainties in absolute sensitivity, as well as in the $\alpha$ and $\beta$ values. One important group currently underrepresented is oxygenated VOCs, including
oxidation products of VOCs. They are highly relevant but challenging to study due to two key issues: the lack of authentic standards and the complexity of these compounds as mixtures. Second, improving the framework will require better knowledge of key thermodynamic properties of analytes. While proton affinity is well-characterized for many compounds, data for $C_6H_6^+$ affinity and the reaction enthalpy of hydride abstraction are scarce. Theoretical calculations offer an efficient means to obtain these values. Third, special cases such as the effects of $O_2$ on the isoprene product ions deserve further attention.
Finally, as instrument conditions (especially declustering that dissociate adduct ions) influence ion chemistry and complicates the product interpretation, caution is required when applying this framework to other instrumental setups.

The benzene CIMS is capable of concurrent detection of NO, VOCs, and their oxidation products. This offers significant advantages by reducing instrumentation complexity and cost while providing collocated, time-resolved measurements of key reactive species. It achieves a 1-minute detection limit of 5 pptv for NO, outperforming most commercial NO$_x$ analyzers. For
many VOCs, the absolute sensitivities and detection limits are comparable to those achieved with state-of-the-art VOCUS PTR reactors. We deployed the benzene CIMS at two sites with contrasting conditions. Isoprene measurements showed good agreement with the GC-PID in St. Louis, but not in Chicago. While many studies have evaluated the instrument's ability to detect isoprene in laboratory settings, field validation remains limited. Our results highlight potential challenges in measuring isoprene in urban environments, where interference from anthropogenic compounds may be significant. NO measurements
from the benzene CIMS showed good agreement with co-located NOx analyzers at both sites. However, due to the limited precision of commercial NOx analyzers, we are unable to rigorously assess the accuracy of the CIMS at low NO concentrations. Future comparisons with more precise instruments would be valuable for evaluating CIMS precision under low-NO conditions. Additionally, better characterizing or mitigating the instrument's humidity-dependent sensitivity would further improve measurement accuracy.

*Data availability.* The field campaign sampling data is available on https://doi.org/10.15485/2547039.



*Author contributions.* UP and LX designed the research, UP operated the CIMS and PICARRO, JRK operated the GC-PID, XC operated the NOx analyzer, and JHC, JL, MAG, and JW provided critical support for the field depolyment. All authors commented on the paper.

*Competing interests.* The contact author has declared that none of the authors has any competing interests.

*Acknowledgements.* We thank Jenna Ditto for sharing the VOC cylinder. We thank Max Berkelhammer from University of Illinois, Chicago
for his coordination of the ground site. We thank Rohan Mishra and John Cavin for useful discussions on the thermodynamics. This work is supported by the U.S. Department of Energy, Office of Science, Office of Biological and Environmental Research, through the Urban Integrated Field Laboratories CROCUS (Collaborative Research on Complex Urban Systems) project under Award Numbers DE-SC0023253 and DE-SC0023226.



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
