# Peer review of "Chemical ionization mass spectrometry utilizing benzene cations for measurements of volatile organic compounds and nitric oxide"

_EGUsphere, 2025_

## Referee Comment (RC2)

**Review of "Chemical ionization mass spectrometry utilizing benzene cations for measurements of volatile organic compounds and nitric oxide"**

This manuscript describes very interesting work on developing benzene cations as a reagent ion for CIMS. The authors demonstrate the surprising and novel result that benzene CIMS can be used to make sensitive measurements of NO in polluted environments. The paper is well written and has several notable strengths. These include the novel ionization energy-based classification, which is a very nice contribution that provides predictive power for work on other compounds. In addition, the two field deployments demonstrate both capabilities (excellent NO measurements) and limitations (isoprene interferences in some urban areas) with a high level of transparency. I highly recommend the paper for publication and would ask the authors to consider the relatively minor points below.

**Line 36** – GC methods are not inherently highly sensitive; it depends on the detector. For example, ECDs are sensitive and the GC provides separation (i.e., selectivity).

**Line 41** – Missing space between "10" and "Hz".

**Line 42** – Maybe instead of saying "ionizes them," replace with "reactions".

**Line 59** – I am not sure a TOF-CIMS with a resolution of 9000 is not more complex than a GC-FID and a NOx box. Certainly bigger and a lot more expensive. I think the time resolution and the ability to detect species without separation are the largest advantages.

**Line 73** – Maybe say "relatively unexplored" instead of "less explored".

**Line 90** – The krypton lamp is referred to as both UV and VUV. I would probably stick to VUV to avoid confusion.

**Line 101** – Reaction R2 is written as an equilibrium reaction, and I think this is correct. So R5 should probably be as well. In fact, the temperature-dependent data show this is true for at least some of the adducts in the SI. I also think it would be reasonable to give an estimate of the thermal lifetime of the benzene cation dimer. This can be done using the equilibrium constant (already reported in the SI) along with an estimate of the forward rate constant for R2 (i.e., Su formulation) to derive a rate constant (and lifetime) for R-2. This could be compared to the residence time in the IMR. I think this would be informative. In addition, the residence time in the IMR is 10 ms; how is this determined? This is not obvious to me. You should also report the pressure in the IMR in the main body, not just in the SI.

**Line 162** – It is a strong statement to say that charge transfer always dominates for low IE even when other channels are available. I would soften that a bit, as I am not sure it will transfer to all molecules and reagent ions.

**Line 164 and other places** – Statements such as "classified into into "mid IE class"" are always a little awkward. Maybe just say you have three classes: Low, Mid, and High IE, and refer to them as "mid IE," for example. Not a big deal, but I trip on these sentences.

**Line 180** – I am not sure I buy the electric field-assisted charge transfer. If you keep it in, you should at least have a reference to this effect in this section.

**Line 188** – The oxygen chemistry of isoprene-derived ions is very interesting (and frustrating for detection). Is there evidence for $O_2^+$ in the mass spectra? If so, that would support it as a driver of the effect but would not be a good sign and would indicate that the ion chemistry is not being controlled well in the ion source. Second, does this type of chemistry happen with other species such as 1,3-butadiene, etc.? These results also beg the question as to why nitrogen was used as the dilution gas when the results in air may be so different. **I think this needs to be addressed and is one of my largest concerns about the work, although I applaud the authors for their transparency.** At a minimum, you need to state what dilution gas is used for all the results (i.e., $H_2O$ dependence, etc.).

**Line 229** – How do you know the output of your $NH_3$ permeation device? The manufacturers only measure mass loss, which may be considerably larger than the emission of $NH_3$ (or other gases). You may be more sensitive than you think.

**Line 235** – You could calculate a collisional rate constant and use your reaction time to estimate a maximum sensitivity.

**Line 309 and later – The humidity dependence is larger than I expected, and I am skeptical it can be fit out to obtain accurate sensitivities.** I appreciate the proposed explanation of the observations (Figure 3). However, I wonder why not run at lower pressures (I am not sure why 55 mbar is considered ideal when you have water effects this large). Perhaps if you halve the IMR pressure, you may see a large decrease in water impacts, as your mechanism indicates it is higher order in water. I would also consider sampling less ambient gas and using more ion source flow or a dilution flow as well. You might sacrifice sensitivity but gain a lot in stability with respect to both water and temperature. **This is my other large concern as to how well this method can be applied.**

*Overall, there is an impressive amount of work and thought detailed in this publication.*

---

## Author Comment (AC1)

We thank the reviewers for their constructive and positive feedback which has helped us improve our manuscript. We respond to the reviewer's comments point by point below. Reviewers' comments are shown in italics. Our responses are shown in plain text, while changes made to the manuscript are shown in blue and enclosed within quotation marks.

**Reviewer #1**

**General Comments**

*Puttu and coauthors present a very comprehensive characterization of chemical ionization mass spectrometry using benzene cluster cation chemistry, highlighting renewed potential utility of this method. Their work not only corroborates and extends prior benzene cluster cation CIMS characterization studies for VOCs but also identifies previously uncharacterized species detectable by benzene CIMS (e.g., NO), which could be highly valuable for future measurements. The methods in the manuscript are detailed and thorough and the presentation is clear and well-structured, with only minor methodological clarifications needed. After addressing these points, I believe this study is well-suited for publication in AMT.*

Authors Reply: We thank the reviewer for the positive feedback on our manuscript.

**Specific Comments**

*1.       Line 97-98: Although larger benzene clusters are not detected because they undergo declustering before reaching the detector, this does not imply that they are absent in the IMR or that they do not influence reagent ion chemistry. While I agree that the key factors determining product ions are limited to ion affinities and IE with respect to the benzene ion monomer or dimer, it should at least be noted that larger clusters may affect reagent ion chemistry unless authors believe that even in the IMR they make up a very small fraction of the reagent ion distribution.*

Authors Reply: It is very unlikely that benzene clusters with n $\geq$3 are present in the IMR under our operating conditions (308K). The equilibrium constant for the trimer (n=3) and tetramer (n=4) formation are $16\times10^{-5}$ Pa$^{-1}$ and $4\times10^{-5}$ Pa$^{-1}$ (Hiraoka et al., 1991). Using $C_6H_6^+/(C_6H_6)_2^+ = 5$, and $P_{C_6H_6}=1.4\times10^{-2}$ Pa, and equilibrium constant equation similar to equation S2, estimated ratios of $(C_6H_6)_3^+/C_6H_6^+$ and $(C_6H_6)_4^+/C_6H_6^+$ are $4.5\times10^{-7}$ and $1.1\times10^{-7}$. Hence benzene cation clusters with n $\geq$3 have negligible abundance in the IMR.

Moreover, we operate RF guides at small voltage gradient ($\Delta V = 1$) and still haven't noticed any benzene clusters with n $\geq$3 in the mass spectrum. Our observation also aligns with a previous study by Lavi et al. (2018) where benzene clusters with n $\geq$3 were not observed, even with the first RF-only quadrupole electronics turned off . Hence it is very unlikely that there are larger clusters in the IMR. We have added this explanation in the main text for further clarity.

"Larger benzene cluster ions $(C_6H_6)_n^+$ with n $\geq$3  likely have negligible abundance in the IMR, because their equilibrium constants for formation are five orders lower than benzene dimer formation (Table S1). This also aligns with Lavi et al., (2018) which showed under the conditions with first radio frequency (RF) only quadrupole electronics turned off, no clusters with size n $\geq$ 3 were observed."

*2.      Line 144-146: This experiment introduces relatively high analyte concentrations. For example, the vapor pressure of alpha-pinene is about 3 torr at room temperature, so dilution gives approximately (3 torr/760 torr) x (0.01 SLPM / 3.01 SLPM) $\approx$ 13 ppm of analyte. Could this affect ion chemistry via titration? Additionally, line 361 notes that higher analyte mixing ratios lead to increased water ion signals, suggesting some effect. Please clarify why the observed product ion distribution is representative of lower analyte concentrations!*

Authors Reply: In the sniff test, the procedure produces an initial burst of analyte with high concentration that temporarily depletes the reagent ions. As the analyte concentration gradually decreased, the reagent ion signal recovered. Figure R1 shows an example time series of analyte and reagent ions in the MACR sniff experiments. Only data collected after the reagent ion signal returned to at least 95% of its baseline value were used in the product ion distribution analysis.

The sniff tests were performed under dry conditions. We have modified the description of the sniff test in the manuscript as below to improve clarity.

"To investigate these ionization pathways, we measured product ion distributions for 27 analytes under dry conditions. In each experiment, 10 sccm of $N_2$ was passed over a vial containing a small amount (< 1 $\mu$L) of pure analyte, and the resulting vapor was diluted with 3 slpm of zero air from a zero air generator (Tisch Environmental, Model ZA-747-30)  before entering the instrument. This procedure leads to an initial burst of analyte that temporarily depletes the reagent ions. As the analyte concentration gradually decreases, the reagent ion signal recovers. Only data collected

after the reagent ion signal returned to at least 95% of its baseline value are used in the product ion distribution analysis. "

[Figure]

**Figure R1.** Time series of MACR signals during its sniff test. The yellow region shows the initial burst of MACR which titrates the reagent ions. This green region, during which the reagent ion signal has recovered, was used for obtaining the product ion distributions.

*3.      Line 185-187: The observed effect of O2 from zero air on the isoprene product ion distribution is interesting. Does a similar distribution occur for other species in the "Mid IE" category? Clarifying whether this behavior is class-specific or species-specific would be helpful.*

Authors Reply: Regarding the $O_2$ effect, we hypothesize that a fraction of the VUV light enters the IMR, leading to the formation of $O_2^+$ and potentially OH radicals. These species could cause complex ionization chemistry and oxidation chemistry of reactive species, thereby contributing to the observed fragmentation. For all of the compounds examined in this study, we observed the $O_2$ effect only for isoprene and limonene. We expect that similar chemistry could also occur for other reactive species, such as butadiene and pentadiene.

Reviewer 2 raised a similar question (i.e., Comment #11). Additional details are provided in our response to that comment.

*4.      Line 344-355: Lavi et al. (2018) observed that for isoprene, the isoprene-benzene cluster signal decreases with increasing humidity, while the charge transfer signal increases, resulting in*

*near conservation of total signal. Is similar behavior observed in these experiments for "mid-IE" species like isoprene or others? If so, how does this reconcile with Figure S10a, where the benzene dimer concentration decreases with increasing water, potentially shifting product ions toward adducts? Since no mechanism is proposed for the sensitivity changes with water (Figure 4), discussing this could help illuminate the ion chemistry.*

Authors Reply: We respectfully disagree with reviewer as Lavi et al. (2018) did not report the behavior claimed in the reviewer's comment for isoprene charge transfer ion. We assume the comment refers to Figure 6 in Lavi et al., (2018) which shows that the water tetramer ion (m/z 73) signal increases with increasing specific humidity. However, this figure does not show the behavior of isoprene charge transfer ion (m/z 68). The observed humidity dependence of the isoprene adduct ion and water tetramer ion is consistent between our study and Lavi et al.

*5.    Line 421: Could potential interferences in the CIMS isoprene signal be verified using the GC measurements? For example, can 1,3-pentadiene be confirmed by GC, and can its ionization energy or ion affinities be compared to assess detectability with benzene CIMS?*

Authors Reply: A GC equipped with a proper column can separate isoprene, 1,3-pentadiene, and other potential isomers. The ionization energy of 1,3-pentadiene (8.6 eV) is very similar to that of isoprene (8.86 eV). Our thermodynamic framework predicts the formation of charge transfer ion for 1,3-pentadiene. We are unable to predict the adduct ion formation because its benzene affinity is currently unknown. While calculating the benzene affinity of 1,3-pentadiene and deploying GC-CIMS are ongoing efforts, the results are not yet ready to be included in this manuscript.

*6.    Figure S8 and SI Lines 138-140: Why does the analyte signal in Fig. S8a rise to a maximum and then decrease? The authors suggest that increased flow raises reagent ion concentration but reduces ion-molecule reaction time, which then reduces product ions but I am a bit confused by this explanation. Since the IMR pressure is held constant, is the residence time in the IMR is primarily not set by its volume and the entrance and exit orifice, so it should not change with reagent ion concentration? If so, could the observed decrease in signal instead reflect shifts in reagent ion chemistry, such as changes in benzene cluster distribution or secondary reactions? Additionally, what does the analyte signal look like beyond the normalized maximum in Fig. S8b? Is there a similar decline that might indicate changes in residence time or reagent ion chemistry? Can you please clarify this in the text and if I am misunderstanding?*

Authors Reply: The residence time in the IMR can be roughly estimated from the ratio of the reactor volume and the total flow rates exiting the IMR. There are two exit flows: one to the IDP3 vacuum pump and the other into the SSQ through an orifice. While the flow rate into SSQ is constant, the flow into the IDP3 pump is regulated by a pressure control valve to maintain the IMR pressure. Under a constant sample flow, which is set by the inlet critical orifice, increasing the reagent ion flow requires the pressure control valve to open further and pump more gas in order to maintain a constant IMR pressure. This increases the total exit flow rate and consequently reduces the residence time. We have revised SI to provide a clearer explanation for the reduction in the residence time.

"…However, at higher flow rates, the benefit of increased reagent ion production becomes smaller, while the higher flow rate reduces residence time in IMR, leading to decreased sensitivity. The residence time in the IMR can be roughly estimated from the ratio of the reactor volume and the total flow rates exiting the IMR. There are two exit flows: one to the IDP3 vacuum pump and the other into the SSQ through an orifice. While the flow rate into SSQ is constant, the flow into the IDP3 pump is regulated by a pressure control valve to maintain the IMR pressure. Under a constant sample flow, which is set by the inlet critical orifice, increasing the reagent ion flow requires the pressure control valve to open further and pump more gas in order to maintain a constant IMR pressure. This increases the total exit flow rate and consequently reduces the residence time."

Figure S8b shows the effect of IMR pressure on analyte sensitivity. In our instrument, the maximum achievable pressure is about 70 mbar when the IDP3 pressure control valve is fully closed. Thus, higher pressure beyond shown in Figure S8b cannot be attained. We expect the sensitivity would continue to increase with increasing pressure due to the higher analyte number density. However, operation at higher pressure may also enhance secondary reactions and degrade detection linearity.

***Technical Corrections***

1.      *Line 42: Consider revising to "The hydronium ion…" or "Hydronium ions are…"*

Authors Reply: We have rephrased this sentence.

"Hydronium ions ($H_3O^+$) are the most widely used reagent ion for measuring VOCs"

**General Comments**

*This manuscript describes very interesting work on developing benzene cations as a reagent ion for CIMS. The authors demonstrate the surprising and novel result that benzene CIMS can be used to make sensitive measurements of NO in polluted environments. The paper is well written and has several notable strengths. These include the novel ionization energy-based classification, which is a very nice contribution that provides predictive power for work on other compounds. In addition, the two field deployments demonstrate both capabilities (excellent NO measurements) and limitations (isoprene interferences in some urban areas) with a high level of transparency. I highly recommend the paper for publication and would ask the authors to consider the relatively minor points below.*

Authors Reply: Dr. Greg Huey, thank you for the positive feedback and insightful suggestions. After submitting the manuscript, we have already begun considering several of the issues you raised and have since conducted some tests. In the response below, we will present some preliminary results as a teaser. Ongoing efforts, including minimizing the isoprene fragmentation, identifying isoprene interferences, and minimizing humidity dependence, will be included in a follow-up manuscript.

**Specific Comments**

*1.     Line 36 – GC methods are not inherently highly sensitive; it depends on the detector. For example, ECDs are sensitive and the GC provides separation (i.e., selectivity).*

Authors Reply: We have revised the sentence as follows

"The GC technique paired with appropriate detectors offer high sensitivity allowing for the detection of numerous VOCs with volume mixing ratios as low as 0.1 pptv (parts-per-trillion by volume). While the GC provides the necessary separation (i.e., selectivity), it provides poorly time-resolved (> 5 mins) measurements. GC typically requires several minutes of sampling to collect sufficient material, followed by additional time to elute analytes through the column."

*2.   Line 41 – Missing space between "10" and "Hz".*

Authors Reply: Done.

3.  *Line 42 – Maybe instead of saying "ionizes them," replace with "reactions".*

Authors Reply: Done

"CIMS selectively detects target species by reactions with reagent ions"

4.  *Line 59 – I am not sure a TOF-CIMS with a resolution of 9000 is not more complex than a GC-FID and a NOx box. Certainly, bigger and a lot more expensive. I think the time resolution and the ability to detect species without separation are the largest advantages.*

Authors Reply: We agree with this comment and have removed the phrase "reduces instrumentation complexity". The revised sentence reads as follows:

"Measurement of both VOCs and NO using a single instrument would be advantageous, as it provides simultaneous, co-sampled, and high time-resolution measurements of key reactive species"

5.  *Line 73 – Maybe say "relatively unexplored" instead of "less explored".*

Authors Reply: Done.

6.  *Line 90 – The krypton lamp is referred to as both UV and VUV. I would probably stick to VUV to avoid confusion.*

Authors Reply: Done.

"Reagent ions are generated by photoionizing benzene using a VUV source (VUV lamp DC PID PKS106, Heraeus)"

7.  *Line 101 – Reaction R2 is written as an equilibrium reaction, and I think this is correct. So R5 should probably be as well. In fact, the temperature-dependent data show this is true for at least some of the adducts in the SI. I also think it would be reasonable to give an estimate of the thermal lifetime of the benzene cation dimer. This can be done using the equilibrium constant (already reported in the SI) along with an estimate of the forward rate constant for R2 (i.e., Su formulation) to derive a rate constant (and lifetime) for R-2. This could be compared to the residence time in the IMR. I think this would be informative. In addition, the residence time in the IMR is 10 ms; how is this determined? This is not obvious to me. You should also report the pressure in the IMR in the main body, not just in the SI.*

Authors Reply: We have revised Reaction R5 to represent it as a reversible reaction.

The equilibrium constant $K_{eq}$ of Reaction R2 is 14 Pa$^{-1}$ under the IMR operation condition (308 K), corresponding to $5.95\times10^{-14}$ cm$^3$ molec$^{-1}$. The forward reaction constant ($k_{forward}$) for Reaction R2 is $2\times10^{-9}$ cm$^3$ molec$^{-1}$ s$^{-1}$, calculated using the parameterization in Su et al. (1994) assuming the reaction proceeds at the collision limit. The reverse rate constant ($k_{reverse}$) is then derived from $k_{forward}$ and $K_{eq}$ as shown in equation S2. The resulting $k_{reverse}$ is $3.36\times10^4$ s$^{-1}$, corresponding to a lifetime of 0.03 ms for benzene dimer ions. Since this lifetime is much shorter than the residence time in the IMR (~10 ms), the benzene monomer ions and dimer ions are in equilibrium.

$$k_{reverse} = \frac{k_{forward}}{K_{eq}} \quad (S2)$$

We have added above calculation in the revised SI.

The 10 ms residence time in the IMR is from Riva et al., which described the design of the reactor used in this study. They calculated the residence time based on the IMR volume and volumetric flows. We have revised the manuscript to accurately reflect the source of the residence time information.

"The residence time in the IMR is approximately 10 milli seconds (Riva et al., 2024), which is calculated based on IMR volume and volumetric flows. "

*8.    Line 162 – It is a strong statement to say that charge transfer always dominates for low IE even when other channels are available. I would soften that a bit, as I am not sure it will transfer to all molecules and reagent ions.*

Authors Reply: We have revised the sentence from "Thus, for analytes in class low IE, charge transfer is the dominant ionization mechanism, regardless of other possible reactions." to the following

"Thus, for analytes in class low IE, charge transfer is likely the dominant ionization mechanism."

*9.    Line 164 and other places – Statements such as "classified into into "mid IE class"" are always a little awkward. Maybe just say you have three classes: Low, Mid, and High IE, and refer to them as "mid IE," for example. Not a big deal, but I trip on these sentences.*

Authors Reply: We have adjusted the phrasing throughout the manuscript and refer to the classes simply as low IE, mid IE, and high IE.

*10.      Line 180 – I am not sure I buy the electric field-assisted charge transfer. If you keep it in, you should at least have a reference to this effect in this section.*

Authors Reply: We agree with the reviewer that this hypothesis may not be correct and have therefore removed this sentence in the revised manuscript.

*11.      Line 188 – The oxygen chemistry of isoprene-derived ions is very interesting (and frustrating for detection). Is there evidence for $O_2^+$ in the mass spectra? If so, that would support it as a driver of the effect but would not be a good sign and would indicate that the ion chemistry is not being controlled well in the ion source. Second, does this type of chemistry happen with other species such as 1,3-butadiene, etc.? These results also beg the question as to why nitrogen was used as the dilution gas when the results in air may be so different.* ***I think this needs to be addressed and is one of my largest concerns about the work, although I applaud the authors for their transparency.*** *At a minimum, you need to state what dilution gas is used for all the results (i.e., $H_2O$ dependence, etc.).*

Authors Reply: Regarding the isoprene fragmentation issue, we hypothesize that a fraction of the VUV light enters the IMR, leading to the formation of $O_2^+$ and potentially OH radicals. These species could cause complex ionization chemistry and oxidation chemistry of reactive species, thereby contributing to the observed fragmentation.

The vendor claims that the VUV lamp is arranged in a way that no VUV light directly enters the IMR. However, this may not be strictly the case. To test this hypothesis, we tuned RF frequency to enhance the transmission of low m/z ions, such as $O_2^+$. Under these conditions, we did observe $O_2^+$ ions and their signal increased with BSQ voltage, reaching a maximum of ~400 cps (Figure R2). We note that $O_2^+$ ions are typically not detected because the RF frequency is normally optimized for higher m/z species, resulting in near-zero transmission efficiency for $O_2^+$.

The observation of $O_2^+$ ions is not entirely unexpected. Your previous study (Ji et al., 2020) reported a similar behavior and proposed a potential solution. Specifically, inserting an elbow between the IMR and the photoionization region can reduce the $O_2^-$ ion signals by 3 orders of

magnitude. We plan to test this configuration in our instrument, but the results will not be available to be included in this manuscript.

[Figure]

**Figure R2.** The $O_2^+$ ion signal during the dry zero air sampling under medium mass mode (green region) and low mass mode (yellow region) at different BSQ ion guide voltages. The white region is the transition period.

In addition to isoprene, similar chemistry is likely to occur for other reactive species, such as butadiene and pentadiene. For most of the compounds examined in this study, the influence of $O_2^+$ is minor; however, we observe similar effects for limonene. This behavior is consistent with our hypothesis that the severity of this issue depends on analyte reactivity. We are conducting additional experiments to further investigate this effect.

Regarding your comment on the dilution gas, nitrogen gas was used only to investigate the isoprene product ion distribution. Zero air was used for all the other analysis, including quantification of

instrument background signal, humidity-dependent sensitivity, and absolute sensitivity. We have revised the manuscript to explicitly state the type of dilution gas used in each specific case.

"Line 145: …was diluted with 3 slpm of zero air before entering the instrument."

"Line 310: ...investigate this, we sample zero air at varying humidity levels."

12. Line 229 – How do you know the output of your $NH_3$ permeation device? The manufacturers only measure mass loss, which may be considerably larger than the emission of $NH_3$ (or other gases). You may be more sensitive than you think.

Authors Reply: We agree with the reviewer that any $NH_3$ loss between the permeation device and CIMS would lead to an underestimation of the reported sensitivity. However, we currently lack a method to quantify the $NH_3$ loss. We considered using ion chromatography, which is a shared instrument, to measure $NH_3$ concentration, but that plan failed due to logistical constraints. We have added the following sentence to acknowledge this limitation.

"For $NH_3$, we use a $NH_3$ wafer device with a permeation rate of $82\pm21$ ng min$^{-1}$ at 40°C (PDWF-0140, VICI Metronics Inc.). Our reported sensitivity should be considered as a lower bound, as any $NH_3$ loss between the permeation device and CIMS would lead to an underestimation of the true sensitivity."

13. Line 309 and later – **The humidity dependence is larger than I expected, and I am skeptical it can be fit out to obtain accurate sensitivities.** I appreciate the proposed explanation of the observations (Figure 3). However, I wonder why not run at lower pressures (I am not sure why 55 mbar is considered ideal when you have water effects this large). Perhaps if you halve the IMR pressure, you may see a large decrease in water impacts, as your mechanism indicates it is higher order in water. I would also consider sampling less ambient gas and using more ion source flow or a dilution flow as well. You might sacrifice sensitivity but gain a lot in stability with respect to both water and temperature. **This is my other large concern as to how well this method can be applied.**

Authors Reply: We thank Dr. Huey for the insightful suggestions. We acknowledge that the IMR pressure used in this study is not optimal for minimizing humidity dependence. The choice of 55 mbar reflects a compromise between absolute sensitivity and humidity-dependent sensitivity. To

be frank, some instrument parameters were selected before we fully understood and quantified the complexities of ion chemistry and humidity-dependent sensitivity. In light of these limitations, our intent is to only report instrument performance under a range of operation conditions without implying any operation conditions as optimal, and to suggest approaches to mitigate humidity dependence for future work.

After manuscript submission, we conducted additional laboratory tests to further address the humidity dependence. One approach we explored was using the water vapor mixing ratio measured by a PICARRO G2401 analyzer, rather than water ion signals measured by the CIMS. As shown in Figure R3, using the PICARRO water removes the influence of analyte concentration on the humidity-dependent sensitivity correction, rendering the proposed 2D interpolation method unnecessary. In cases where an external water vapor measurement is not available, we also tested the use of an internal calibrant (e.g., bromoanisole) to track changes in instrument performance and the water vapor mixing ratio in the sample.

As suggested by the reviewer, operating at a lower IMR pressure and introducing additional dilution flow can reduce humidity dependence. Optimizing the instrument configuration requires substantial effort. We also plan to conduct a field campaign this summer to further evaluate these approaches under ambient conditions.

To avoid overloading the current manuscript, we prefer not to include these additional tests here. Instead, we have added a discussion in the revised manuscript to describe potential mitigation strategies for future work.

"Additionally, better characterization and mitigation of the instrument's humidity-dependent sensitivity would further improve measurement accuracy. For example, using an external water vapor measurement rather than water ion signals measured by the CIMS could remove the influence of analyte concentration on the humidity-dependent sensitivity correction. Operating at lower IMR pressure or introducing dry dilution gas could also mitigate humidity dependence. Taking NO as an example, diluting an ambient sample with a 2% water mixing ratio by a factor of 4 to reduce the water mixing ratio to 0.5% could increase the sensitivity by a factor of 8 (Figure 4), thereby compensating for the decrease in absolute sensitivity due to dilution. These approaches will be the focus of our future work."

[Figure]

**Figure R3.** Toluene signal (normalized to dry signal) as a function of (a) water vapor signal in CIMS (water tetramer ion, $H_9O_4^+$) (b) $H_2O$ mixing ratio as measured by PICARRO for 3 different mixing ratios of toluene (3, 5, and 8 ppbv).

**Additional changes**

1. We have added Bingru Wang as a co-author because he provided the dataset used to quantify the transmission efficiency. This contribution was inadvertently omitted in the initial submission and has now been included in the "Author Contributions" section.

2. After submitting the manuscript, we realized that our algorithm to classify product ions into specific reaction pathways (Section S3.1 of the original SI) is not accurate. The primary limitation arises from distinguishing between adduction ion and charge transfer ion, as a given molecular formula may correspond to both ionization pathways.

We have updated the discussion and Figure 10 in the revised manuscript as follows:

"We classify the detected ions into two groups based on whether their degree of unsaturation is an integer: (1) adduct and charge transfer ions (integer values) and (2) proton transfer and hydride abstraction ions. Further classification of ionization pathways based solely on ion formula is challenging, as a given molecular formula may correspond to multiple ionization pathways. For example, $C_7H_8O^+$ may arise from charge transfer of cresol or from adduct formation with

formaldehyde. Although such ambiguities exist, ionization pathways can be inferred using prior knowledge of analyte properties or laboratory experiments. In this case, $C_7H_8O^+$ is likely the charge transfer of cresol, as benzene CIMS is not sensitive to formaldehyde based on laboratory experiments. Among all detected ions, adduct and charge transfer ions account for 49%, with the reminder consisting of proton transfer and hydride abstraction ions. Within the adduct/charge transfer category, 65% of the ions are $C_xH_yO_z^+$ and 26% are $C_xH_y^+$. Within proton transfer/hydride abstraction category, 64% are $C_xH_yO_z^+$ and 17% are $C_xH_y^+$."

**References:**

Hiraoka, K., Fujimaki, S., Aruga, K., and Yamabe, S.: Stability and structure of benzene dimer cation (C6H6)+2 in the gas phase, J. Chem. Phys., 95, 8413–8418, https://doi.org/10.1063/1.461270, 1991.

Ji, Y., Huey, L. G., Tanner, D. J., Lee, Y. R., Veres, P. R., Neuman, J. A., Wang, Y., and Wang, X.: A vacuum ultraviolet ion source (VUV-IS) for iodide–chemical ionization mass spectrometry: a substitute for radioactive ion sources, Atmospheric Measurement Techniques, 13, 3683–3696, https://doi.org/10.5194/amt-13-3683-2020, 2020.

Lavi, A., Vermeuel, M. P., Novak, G. A., and Bertram, T. H.: The sensitivity of benzene cluster cation chemical ionization mass spectrometry to select biogenic terpenes, Atmospheric Measurement Techniques, 11, 3251–3262, https://doi.org/10.5194/amt-11-3251-2018, 2018.

Meot-Ner, M., Hamlet, P., Hunter, E. P., and Field, F. H.: Bonding energies in association ions of aromatic compounds. Correlations with ionization energies, J. Am. Chem. Soc., 100, 5466–5471, https://doi.org/10.1021/ja00485a034, 1978.

Riva, M., Pospisilova, V., Frege, C., Perrier, S., Bansal, P., Jorga, S., Sturm, P., Thornton, J. A., Rohner, U., and Lopez-Hilfiker, F.: Evaluation of a reduced-pressure chemical ion reactor utilizing adduct ionization for the detection of gaseous organic and inorganic species, Atmospheric Measurement Techniques, 17, 5887–5901, https://doi.org/10.5194/amt-17-5887-2024, 2024.

Su, T.: Parametrization of kinetic energy dependences of ion–polar molecule collision rate constants by trajectory calculations, The Journal of Chemical Physics, 100, 4703, https://doi.org/10.1063/1.466255, 1994.